



# A multi-disciplinary analysis of the exceptional flood event of July 2021 in central Europe. Part 2: Historical context and relation to climate change

Patrick Ludwig[1,2], Florian Ehmele[2], Mário J. Franca[3], Susanna Mohr[1,2], Alberto Caldas-Alvarez[2], James E. Daniell[1,4], Uwe Ehret[1,3], Hendrik Feldmann[2], Marie Hundhausen[2], Peter Knippertz[2], Katharina Küpfer[1,2], Michael Kunz[1,2], Bernhard Mühr[1], Joaquim G. Pinto[1,2], Julian Quinting[2], Andreas M. Schäfer[1,5], Frank Seidel[3], and Christina Wisotzky[1,6]

[1]Center for Disaster Management and Risk Reduction Technology (CEDIM), Karlsruhe Institute of Technology (KIT), Karlsruhe, Germany
[2]Institute of Meteorology and Climate Research, Karlsruhe Institute of Technology (KIT), Karlsruhe, Germany
[3]Institute for Water and River Basin Management, Karlsruhe Institute of Technology (KIT), Karlsruhe, Germany
[4]Institute of Photogrammetry and Remote Sensing, Karlsruhe Institute of Technology (KIT), Karlsruhe, Germany
[5]Geophysical Institute, Karlsruhe Institute of Technology (KIT), Karlsruhe, Germany
[6]Institute of Economics, Karlsruhe Institute of Technology (KIT), Karlsruhe, Germany

**Correspondence:** Patrick Ludwig (patrick.ludwig@kit.edu)

**Abstract.** Heavy precipitation over western Germany and neighboring countries in July 2021 led to widespread floods, with the Ahr and Erft river catchments being particularly affected. Following the event characterization and process analysis in Part 1, here we put the 2021 event in the historical context regarding precipitation and discharge records, and in terms of the temporal transformation of the valley morphology. Furthermore, we evaluated the role of ongoing and future climate change
on the modification of rainfall totals and associated flood hazards as well as implications for flood management.

The event was among the five heaviest precipitation events of the past 70 years in Germany. However, considering the large LAERTES-EU regional climate model (RCM) ensemble revealed a substantial underestimation of return values and periods based on extreme value statistics using only observations. An analysis of homogeneous hydrological data of the last 70 years demonstrated that the event discharges exceeded by far the statistical 100-year return values. Nevertheless, the flood peaks at
the Ahr River were comparable to the reconstructed major historical events of 1804 and 1910, which were not included in the hazard assessment of flood risk so far. A comparison between the 2021 and past events showed differences in terms of the observed hydro-morphodynamic processes which enhanced the flood risk due to changes in the landscape organization and occupation.

The role of climate change and how the 2021 event would unfold under warmer or colder conditions (within a $-2\,$K to
$+4\,$K range) was analyzed based on pseudo-global-warming (PGW) model experiments. These showed that the spatial mean precipitation scales to first order with the theoretical Clausius-Clapeyron (CC) relation predicting a 7 to 9 % increase per degree warming. Using the PGW rainfall simulations as input to a hydrological model of the Ahr river basin revealed a strong and non-linear effect on flood peaks: For the $+2\,$K scenario, the 18 % increase in areal rainfall led to a 39 % increase of the flood peak at gauge Altenahr. The analysis of the high-resolution convection-permitting KIT-KLIWA RCM ensemble confirmed



the CC-scaling for moderate spatial mean precipitation but showed a super CC-scaling of up to 10 % for higher intensities. Moreover, also the spatial extent of such precipitation events is expected to increase.

## 1  Introduction

In mid-July 2021, heavy precipitation over two days exceeding 150 mm affected a large area covering western Germany with the adjacent regions in the Netherlands, Belgium, Luxembourg, and France, triggering widespread flooding (e.g., Schäfer et al.,

2021; Junghänel et al., 2021; Dewals et al., 2021; MeteoLux, 2021). Especially areas in the German federal states of North-Rhine Westphalia (NRW) and Rhineland-Palatinate (RP), Luxembourg, and eastern Belgium were heavily affected by floods, in particular, the river catchments of the Ahr and Erft (Fig. 1a). The area of these two is characterized by the low mountain ranges of the Eifel and the Ardennes. Both the Ahr and the Erft Rivers are tributaries of the Rhine River, the former with a catchment area of approx. $900\,\mathrm{km}^2$, and the latter with approx. $1800\,\mathrm{km}^2$. On 14 July 2021, precipitation totals widespread

reached values of more than 75 mm in 24 hours (locally even over 100 mm; Fig. 1b) with most of the precipitation even falling within 15 hours (Mohr et al., 2022). Severe damage to buildings, infrastructure, and industry as well as the loss of over 180 lives was the result. The event was one of the five costliest disasters in Europe in the last half-century (Mohr et al., 2022), with Munich Re (2022) estimating a total loss of 46 billion Euros and 33 billion Euros in Germany alone. Even one year later, reconstruction work is still ongoing, and it will take years until all infrastructure is back in place (BMDV, 2021, 2022).

This two-part interdisciplinary study emerged from activities (Schäfer et al., 2021) within the Center for Disaster Management and Risk Reduction Technology (CEDIM; www.cedim.kit.edu; last access: 9 May 2022), an interdisciplinary research center in the field of disasters, risks, and security at the Karlsruhe Institute of Technology (KIT), Germany. CEDIM conducts so-called Forensic Disaster Analyses (FDA) in near-real-time since 2011 (e.g., Kunz et al., 2013; Merz et al., 2014; Piper et al., 2016; Wilhelm et al., 2021) aiming to get an overview of a disaster including identification of main drivers and assessing

related impacts within a few hours to days after the event. The first part of this study (Mohr et al., 2022, hereafter referred to as PART1) focused on the characterization and analysis of the event itself encompassing the interlink of meteorological, hydrological, and hydro-morphological processes and effects. In addition, the extension of the inundation areas and impacts like traffic disruption and economic losses were addressed.

The synoptic large-scale conditions that led to the heavy precipitation event were characterized by a quasi-stationary, large-

scale trough and an associated low-pressure system over the region, which was sustained by a blocking event over the eastern North Atlantic. These synoptic patterns were well predicted by the numerical weather prediction models operated by Deutscher Wetterdienst (DWD) or the European Centre for Medium-Range Weather Forecasts (ECMWF). Such large-scale situations foster extreme weather events like the one discussed here (e.g., Woollings et al., 2018; Kautz et al., 2022). The predicted and observed rainfall totals were extreme for that particular region exceeding high return periods affecting an uncommonly large

area (cf. PART1). The recorded and reconstructed flood peaks were extraordinary and were exacerbated by the morphological characteristics of the catchments and river channel network, and by the landscape occupation and organization, both responsible for widespread inundation of the valley, the generalized occurrence of erosion and scouring processes, deposition, clogging,





and damming of channel network bottlenecks such as bridges, streets, and narrow river sections, or the collapse of the structure of the channel network with the observation of flow bypasses and riverbank collapses.

A key aspect that needs deeper analysis is the evaluation of the 2021 flood event from a long-term climatological perspective. For example, the Ahr catchment had been affected by two severe flood events in 1804 and 1910 (Roggenkamp and Herget, 2014a, b). In spite of the limited data available, evidence is given that these events might have been comparable to the July 2021 event in terms of discharge (PART1; Roggenkamp and Herget, 2022). However, these events were not considered for the estimation of the 100-year return periods of discharge ($HQ_{100}$) as the continuous time series of observations only starts in
1946. Thus, Vorogushyn et al. (2022, in review) recently estimated the return period of the 2021 flood at gauge Altenahr (Ahr) based only on the recorded data from 1946 to 2019 to be more than $10^8$ years, which is very unrealistic and clearly shows the limits of the extreme value statistics for rare events. In contrast, taking into account reconstructed historical floods since 1804, the return period of the 2021 flood is reduced to an order of magnitude of $HQ_{10\,000}$. In addition, the consequences such as inundation areas and depths in the valley during the past and the 2021 floods differ dramatically in some cases. This can
also be attributed to changes in the landscape or in landscape use and newly emerging process connections and feedbacks (cf. PART1; Dietze et al., 2022).

Because the dimensions of the July 2021 event were somehow surprising and unexpected, the role of ongoing and future climate change (IPCC, 2021, 2022) on the evolution and characteristics of such extreme events is an important issue to be addressed. According to the Clausius-Clapeyron (CC) relationship, the intensity and probability of precipitation events are
affected by an increase in moisture content of 7 % per degree temperature increase (e.g., Allen and Ingram, 2002; Pall et al., 2007). In fact, on the global scale, the general circulation models (GCMs) project an increase in heavy precipitation over large areas of the globe which, however, scales not necessarily with the CC-rate (Stocker, 2014; IPCC, 2021). One possible reason is the resolution of GCMs (usually 100 to 200 km), which is too coarse to fully capture the local characteristics and intensities of (heavy) precipitation events contributing to the total amounts. A possible approach to overcome this shortcoming is the
consideration of regional climate model (RCM) simulations with typical horizontal resolutions of 12 to 25 km or even down to convection-permitting resolutions below 3 km (e.g., Vergara-Temprado et al., 2021). A good overview of the advantages of large RCM ensembles is given in Maher et al. (2021). For example, for the historical period, the LAERTES-EU data set (Ehmele et al., 2020, 2022) with over 12 000 years at 25 km horizontal resolution provides an excellent basis to estimate how uncommon precipitation values are in the scope of recent climate conditions. Moreover, the examination of convection-
permitting ensemble simulations (Prein et al., 2015; Ban et al., 2021) further improves the basis for such an evaluation, given that such models exhibit a largely reduced bias in terms of precipitation intensities compared to lower resolution (12 to 25 km) climate models (Prein et al., 2015; Caldas-Alvarez et al., 2022b).

While conventional climate model simulations are used to assess the general evolution of precipitation characteristics, so-called attribution studies elaborate on how climate change affects specific events. Such studies consider a very large number of
climate model simulations with and without anthropogenic forcing (probabilistic event attribution). By comparing the statistics of both types of simulation, it is possible to estimate whether the probability of occurrence of a specific event has changed in recent decades or not (Allen and Ingram, 2002; Stott et al., 2004, 2016; Otto, 2017). Furthermore, climate model simulations

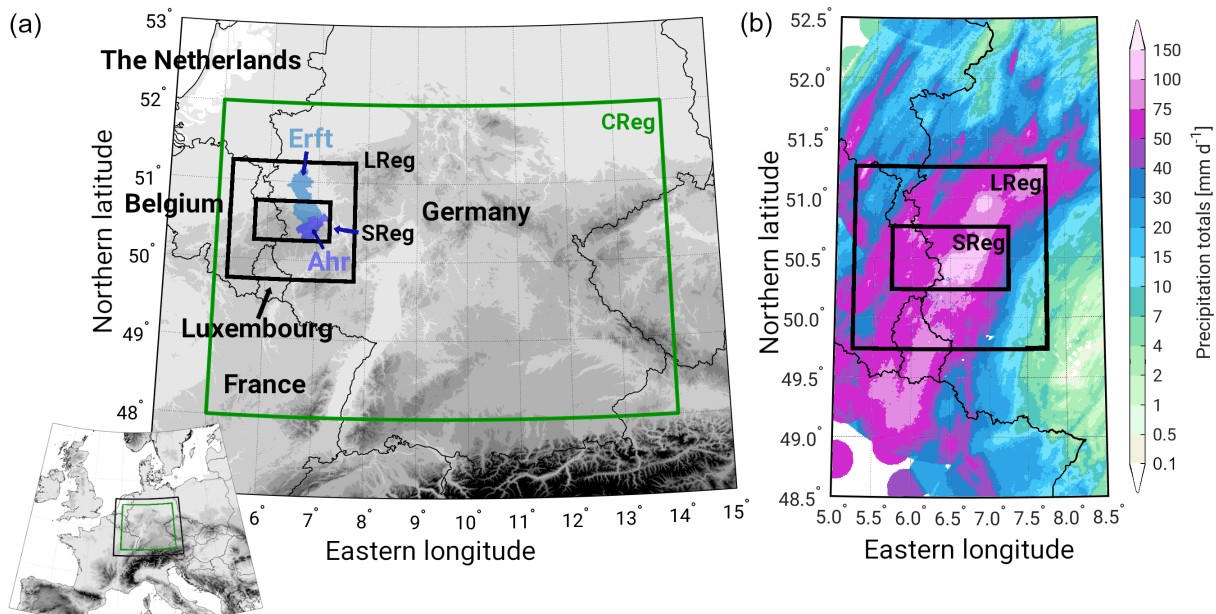

**Figure 1.** (a) Overview map of central and western Europe (bottom left) with a zoom-in on the region of interest, and (b) 24-hour precipitation totals of the 14 July 2021 based on RADOLAN (cf. PART1). Two regions (LReg, SReg; black rectangles) covering the main precipitation area and the affected river catchments of the Ahr (dark blue) and the Erft (light blue) are defined to derive specific event characteristics (e.g., spatial means). Analyses in a statistical and climatological context are performed over the greater central Germany region (CReg, green box). Topographic data (grey) were provided by NOAA National Geophysical Data Center (Amante and Eakins, 2008).

can also be used for conditional event attribution investigations as an alternative to the above-described probabilistic approach (Stott et al., 2016). The conditional approach is often referred to as "storyline" approach (e.g., Shepherd, 2016; Shepherd

et al., 2018; Shepherd, 2019), which assesses the extent to which recent climate change and future projected conditions could affect the magnitude of a specified event (e.g., Trenberth et al., 2015; Sillmann et al., 2021; Sánchez-Benítez et al., 2022) by performing coupled climate model simulations. A variant of this approach is to assess climate change and its impact on extreme events with a pseudo-global-warming (PGW) analysis (e.g., Schär et al., 1996; Michaelis et al., 2017). In this case, the thermodynamic modifications are imprinted to the initial forcing data of RCM simulations, such as temperature changes

corresponding to a fixed warming level (e.g., +2 K) or by considering the mean changes from the background environmental conditions from GCMs under a given scenario. For the July 2021 event, a study from the World Weather Attribution initiative (e.g., Otto, 2017) estimated an increase in the probability of occurrence already by a factor between 1.2 and 9 compared to a 1.2 K colder climate (Kreienkamp et al., 2021). Examples of similar work have already been performed for other severe flood events (e.g., Lackmann, 2013; Lenderink et al., 2021).

In the present study (hereafter referred to as PART2), we put the July 2021 event into a historical context using both observational and the LAERTES-EU data sets. Regarding climate change, we consider an ensemble of convection-permitting climate





simulations and novel PGW simulations that were performed specifically for this event. The following research questions are addressed:

(I) How does the event classify within the historical context of precipitation and flood events?

(II) In which way did the historical transformation of river valleys (e.g., landscape occupation and organization) change the 2021 flood hazard in comparison to past events in this region?

(III) How would the July 2021 precipitation event unfold under different past and future climatic conditions and what implications have these scenarios on flood events?

(IV) How are precipitation characteristics (e.g., intensity, extent) projected to change under future climate conditions?

The structure of PART2 is as follows: the data sets, models, and methods are described in Sect. 2. The classification into the historical context is presented in Sect. 3. Section 4 focuses on the possible role of climate change for the event and future projections. Finally, the discussion, summary, conclusions, and outlook are presented in Sect. 5.

## 2 Data and methods

In PART2, three rectangular boxes are defined for both the event characterization and the statistical analysis. For a long-term 115 climatological and statistical analysis, we define the CReg area (green rectangular in Fig. 1a) covering central Germany and parts of neighboring countries between $5°$ E and $14°$ E, and between $48°$ N and $52°$ N with about $285\,000\,\text{km}^2$. Although the July 2021 event primarily affected western Germany, our intention is to identify comparable heavy precipitation events across Germany in terms of spatio-temporal extent, precipitation totals, and antecedent conditions. CReg is characterized by similar topographic features such as low-mountain ranges resulting in similar orographic forcing during precipitation formation or 120 intensification. Furthermore, CReg is a region with hydroclimatic conditions comparable to the Ahr and Erft river basins such that analyses of both extreme rainfall and flood events can serve as a comprehensive spatio-temporal context for the 2021 flood event.

To characterize the July 2021 event more specifically, two additional smaller regions are defined based on the precipitation observations and most affected areas of the Ahr and Erft river catchments: The larger event area (LReg, see Fig. 1) ranges 125 between $5.25°$ E and $7.75°$ E, and from $49.75°$ N to $51.25°$ N covering an area of approx. $30\,000\,\text{km}^2$. The smaller event area (SReg) ranges between $5.75°$ E and $7.25°$ E, and from $50.25°$ N to $50.75°$ N with a covered area of approx. $6000\,\text{km}^2$.

The investigated period, in general, is determined by the length of the used data sets (see below). For the historical classification, all available data are used, the July 2021 event itself is characterized temporally using the 24 hours from 14 July 2021 05:50 UTC to 15 July 2021 05:50 UTC.



## 2.1 Observational data

### 2.1.1 Precipitation data

In line with PART1, two different gridded precipitation data sets provided by Deutscher Wetterdienst (DWD) are used in this study: daily HYRAS data (*Hydrometeorologische Rasterdatensätze*; Rauthe et al., 2013), and hourly RADOLAN data (*Radar-Online-Aneichung*; Weigl and Winterrath, 2009; Winterrath et al., 2018). HYRAS includes daily precipitation totals at a $5 \times 5$ km$^2$ grid resolution covering Germany and its relevant river basin in neighboring countries for the period from 1951 to 2015 (update in preparation by DWD). HYRAS is based on station measurements interpolated to the regular grid considering local characteristics such as elevation or exposition. A sub-sample of HYRAS are the HYRAS-DE data, formerly known as REGNIE (*Regionalisierte Niederschlagshöhen*), covering only Germany but with a higher resolution of 1 km$^2$ and continuous updates on a daily basis. While HYRAS with its larger spatial extent was used for the bias correction of the high-resolution regional climate simulations (see Sect. 2.2.1), HYRAS-DE is used to find comparable historical events in Germany due to the longer time period covered.

RADOLAN is a radar-based near-real-time precipitation data set covering Germany and parts of the neighboring countries with roughly 1 km$^2$ horizontal and hourly temporal resolution available since 2001. To account for uncertainties and typical radar artifacts, the radar-based precipitation rates are calibrated using hourly data of over 1000 ground-based observational stations. RADOLAN is used to derive the precipitation totals of the July 2021 event due to its spatio-temporal availability.

### 2.1.2 River gauge data

As described in detail in PART1, we collected water level and streamflow data of ten river gauges for the 2021 flood event, covering the study area LReg (see Fig. 1) from the river Wupper in the east to the river Prüm in the west. These gauges cover a range of basin sizes from 31.9 km$^2$ at gauge Schönau (river Erft) to 816 km$^2$ at gauge Kordel (river Kyll). The gauge locations are shown in PART1, Figure 1, water level and streamflow time series during the 2021 event are shown in PART1 (Fig. 5). Further information such as basin size, historical extremes, and the discharge estimates of the 2021 flood event are also listed in PART1 (Table 1). Henceforth, we refer to these data as 2021 gauge data (2021GD). All data were provided by the water administrations of Rhineland-Palatinate, the Erftverband, and the Wupperverband.

To put the 2021 flood into a broader historical context, we utilized three further hydrological data sets. The first is a collection of all streamflow data in the greater central Germany region (CReg, see Fig. 1) available from the Global Runoff Data Center (GRDC). To ensure comparability with the 2021GD, we restricted our sample to basin sizes up to 1000 km$^2$. The 124 gauge time series available from GRDC fulfilling this criterion cover on average 69 years adding up to 9799 years of observations in total. From each time series of mean daily streamflow in m$^3$ s$^{-1}$, we extracted the maximum value, i.e., the highest flood on record for each gauge. These maxima serve as an empirical upper bound for peak streamflow as a function of basin size. Henceforth, we refer to this data set as GRDC data.

The second data set was taken into account to classify the peak flows observed in 2021 in terms of statistical return periods. As a reference, we used peak discharge magnitudes for statistical return periods of 100, 200, 500, 1000, 5000, and 10 000 years





(henceforth HQ$_{100}$, HQ$_{200}$, etc.) provided by the *Landesanstalt für Umwelt Baden-Württemberg* (LUBW) for all river gauges in the federal state of Baden-Württemberg. Ideally, such gauge data should have been used from the entire CReg region rather than

from Baden-Württemberg only. However, the CReg region extends over several federal states, and as the water administration in Germany is under the responsibility of the federal states, it is impossible to gather flood return times for all gauges in the CReg region based on the same approach of extreme value statistics. As Baden-Württemberg has a large spatial overlap with the CReg region, we considered this data set, comprising overall 355 gauges as a suitable basis to derive robust estimates of the relation between the magnitude of the 100-year flood and floods of higher return periods. We will refer to this data set

as LUBW data in the text. Please note that while the GRDC data set is based on daily averaged data, the LUBW data are based on hourly data. Therefore, the 2021GD is used as daily averages for comparison with the former and as hourly data for comparison with the latter. As mentioned in PART1, we do not attempt to assign a particular return period to the event as the related uncertainties are very large. Nevertheless, we think there is an added value in providing a broader classification using the LUBW and GRDC data.

The third data set contains peak discharge values of major floods between 1804 and 2021 at gauge Altenahr (Ahr) – one of the basins most severely affected by the 2021 flood (Roggenkamp and Herget, 2022). The data are based on gauge recordings since 1946, and on reconstructions before. We use this data set to put the 2021 flood at the Ahr River into a larger historical perspective of local floods.

## 2.2   Model simulations

### 180   2.2.1   The regional climate model COSMO-CLM

Two ensemble data sets using the non-hydrostatic model of the Consortium for Small-scale Modeling (COSMO) in climate mode COSMO-CLM (CCLM; Sørland et al., 2021; Baldauf et al., 2011) were considered. One of them, the LAERTES-EU large regional ensemble (Ehmele et al., 2020) consists of CCLM simulations at a resolution of 0.22° (≈ 25 km). The simulations were performed within the MiKlip project (Marotzke et al., 2016), which developed an operational decadal prediction system

based on the Max Planck Institute of Meteorology coupled Earth System Model (MPI-ESM) with a regional downscaling component (Feldmann et al., 2019). Several ensemble generations of initialized decadal hindcast simulations with a consistent model chain were combined into the LAERTES-EU ensemble, which consists of about 12 500 simulation years covering the present-day climate (the 20th century and the beginning of the 21st century). A positive precipitation bias of LAERTES-EU compared to observations was identified by Ehmele et al. (2020), hence, Ehmele et al. (2022) applied a bias correction

via monthly quantile mapping using E-OBS (Haylock et al., 2008) as a reference, which reduced the bias significantly. By applying the bias-corrected LAERTES-EU ensemble to hydrologic modeling of major Central European river basins, Ehmele et al. (2022) showed that LAERTES-EU enables statistically robust estimations of extreme events with very high return periods.

   To assess the effect of climate change on extreme precipitation intensities and their return values over complex topography, very high-resolution climate simulations are needed (Feldmann et al., 2013; Prein et al., 2015). For this purpose, an ensemble

of regional climate simulations with CCLM at convection-permitting (CPM) resolution of 2.8 km was produced and applied in





the context of the KLIWA (*Klimaveränderung und Wasserwirtschaft*) project (Schädler et al., 2018; Hackenbruch et al., 2016) at KIT (hereafter referred to as KIT-KLIWA). The KIT-KLIWA ensemble consists of transient simulations covering the time period from 1971 to 2100. Four general circulation models (GCMs) from the Coupled Model Intercomparison Project Phase 5 (CMIP5) (Taylor et al., 2012) using the Representative Concentration Pathway emission scenario 8.5 (Meinshausen et al., 2011,

RCP8.5;) provided the boundary conditions for a three-step downscaling with CCLM, first to 50 km over Europe, second to 7 km over Germany, and finally to 2.8 km over Southern Germany south of 52° N. Subsequently, a bias correction via monthly quantile mapping (Berg et al., 2012) was applied to the daily precipitation totals using the HYRAS data as a reference. An overview of the driving GCMs and realizations used for KIT-KLIWA can be found in Table S1 in the supplementary material. The KIT-KLIWA simulations are analyzed regarding the extreme precipitation in a present-day reference period (1971 to 2000)

and their changes under a global warming level (GWL) of +2 K and +3 K (hereafter GWL2 and GWL3) with respect to pre-industrial (1881 to 1910) climate conditions. The method was adopted from Teichmann et al. (2018), who defined the GWL as a 30-year period centered around the year in which a GCM reaches the GWL for the first time. Consequently, this time period varies between the different GCMs used, but as a result, they represent similar climatic conditions. The GWL time periods in the respective GCMs can be found in Table S1. The GWL of the present-day reference period based on global observations is

+0.46 K (Teichmann et al., 2018). Note that the GWL refers to a global average and that regional warming levels might deviate from this.

### 2.2.2   Pseudo-global-warming experiments with WRF

To assess how the flood event could potentially unfold in the context of climate change, a storyline approach is applied (Shepherd, 2016). This approach complements the above-described probabilistic concept based on ensembles of (regional) climate

model simulations to cope with the uncertainty in physical aspects of climate change. With that, a series of PGW experiments with the WRF model (version 4.3; Skamarock et al., 2019) were performed. A control simulation at a CPM horizontal grid spacing of 0.0275° ($\approx$ 2.8 km) driven by the fifth generation European Centre for Medium-Range Weather Forecasts (ECMWF) atmospheric reanalysis data (ERA5; Hersbach et al., 2020) was conducted to test the capability of the WRF model to simulate the precipitation event appropriately. All WRF simulations were initialized on 14 July 2021, 00:00 UTC and ran for 30 hours

until 15 July 2021, 06:00 UTC with boundary conditions being updated hourly. To ensure that the simulated flood triggering cyclone remains over the affected area, spectral nudging (von Storch et al., 2000) was applied to the large-scale wind fields above the planetary boundary layer. To account for the physical processes that are not explicitly resolved by the model, we used the following parameterization schemes: the rapid radiative transfer model RRTMG for shortwave and longwave radiations (Iacono et al., 2008), the Thompson microphysics scheme (Thompson et al., 2008), Zhang et al. (2011), the Mellor-Yamada-Janjic

(MYJ) planetary boundary layer scheme (Janjić, 1994), the Noah land surface model (Chen et al., 1997), and the MYJ surface layer scheme (Janjić, 1994). For the PGW experiments, the control simulation was repeated with changes in the ERA5 initial and boundary conditions. In total, we performed ten additional PGW experiments, with the temperatures of the initial and boundary data from ERA5 being either reduced to –2 K or increased to +3 K at an interval of 0.5 K. A reduction of 1 K would represent temperatures as in the pre-industrial period, while an increase of 2 K corresponds to GWL3. The relative humidity





was kept constant by reducing or increasing the specific humidity of the initial and boundary data based on the Clausius-Clapeyron (CC) relationship, which describes that for every 1 degree increase in temperature there is an increase in humidity of 7 %.

### 2.2.3 Hydrological simulations with LARSIM

The results of the PGW simulations described in the previous section are used for additional hydrological modeling and
analyses. Therefore, the operational flood forecasting model based on the hydrological Large Area Runoff Simulation Model (LARSIM) (Ludwig and Bremicker, 2006) for RP is used, which is operated by the water administration of RP. LARSIM is a semi-distributed, physics-based conceptual water balance model that captures all relevant processes related to the terrestrial water cycle and operates in hourly resolution. In Germany, it is in widespread use for operational flood forecasting and water balance modeling. For the PGW-based studies, the LARSIM model is used for the Ahr river upstream of gauge Altenahr
(hereafter LARSIM-Ahr). The LARSIM-Ahr model consists of 561 subbasins with an average area of $1.6\,\mathrm{km}^2$ per subbasin.

Three different simulations are performed with LARSIM-Ahr. The first run was forced with a spatially distributed rainfall product as presented by Bardossy et al. (2022). It is based on a comprehensive post-event collected set of rain gauge recordings from both public and private weather stations, and rainfall estimates based on the signal attenuation in mobile phone networks. A comparison with radar-based rainfall estimates and estimates from public rain gauges only (Regenauer et al., 2022) demon-
strated that this product allowed the most accurate simulation of the reconstructed flood peak at Altenahr. Hence, this reference was used as a basis for the hydrological PGW experiments. For the other two LARSIM-Ahr simulations, the PGW relations for $-1\,\mathrm{K}$ (pre-industrial) and $+2\,\mathrm{K}$ (GWL3) are used.

### 2.3 Methods

#### 2.3.1 Return values and periods

For both observations and model data, the return period (RP) $T_{\mathrm{RP}}(x)$ of a precipitation event $x$ or, vice versa, the return value (RV) for a given return period $x_{\mathrm{RV}}(T)$ is derived using extreme value statistics. In line with PART1, we used the three-parameter generalized Pareto distribution (GPD) and applied it to the KIT-KLIWA climate model ensemble (see Sect. 2.2.1) using the peaks-over-threshold (POT) approach (e.g., Wilks, 2006). The scale and shape parameters of the GPD were estimated using maximum likelihood estimation (MLE), while the 95 % percentile of the considered data series was used as the location
parameter. After estimating the parameters, the statistical relation between the cumulative density function of the GPD $F_{\mathrm{GPD}}$ and the corresponding return period $T$ can be expressed as $T = [\lambda \cdot (1 - F_{\mathrm{GPD}})]^{-1}$ (e.g., Madsen et al., 1997; Brabson and Palutikof, 2000), where $\lambda$ is the crossing rate (average number of events per year).

Adjusting such a statistical distribution function to a data series reduces the statistical uncertainty compared to an empirical return period estimation such as the block maximum method (e.g. Bezak et al., 2014). Furthermore, it allows for extrapolating
the analyses to higher return periods beyond the length of the given time series which is useful for comparatively short data sets like observations (e.g., Grieser et al., 2007). However, the uncertainty of this extrapolation rapidly increases in this uncovered





range of values. Following the study of Früh et al. (2010), who found reasonable results of extreme value statistics for return periods of one-third of the length of the time series, we focus on the 10-year return value in the case of KIT-KLIWA.

In contrast to adjusting a statistical distribution, the empirical approach was applied to the LAERTES-EU data to determine
return values and periods up to 1000 years, which is adequate due to the large data amount of about 12 500 years. In this case, the return period estimation simplifies to $T = L \cdot N^{-1}$, where $L$ is the length of the data series and $N$ is the number of occurrences (Gumbel, 1941).

### 2.3.2 Precipitation indices

In addition to the antecedent precipitation index (API) used in PART1, two other measures are applied to the HYRAS-DE data
for the classification of precipitation events: the first one is the empirical heavy precipitation event criterion HPE$_{\text{crit}}$, which combines thresholds for magnitude and extension. A special feature of the July 2021 event was the rather wide area with high precipitation totals exceeding the 50-year return level (cf. PART1, Fig. S3), which, in combination with the hydrological and hydro-morphological conditions on-site, resulted in this exceptional event. Therefore, an event fulfilling HPE$_{\text{crit}}$ is defined when daily rainfall totals exceed the 50-year return level according to KOSTRA (cf. PART1, supplementary material) on a
contiguous area $A$ of at least $1000\,\text{km}^2$, representing roughly medium-size river catchments like the Ahr and the Erft. Thus, comparable precipitation events in terms of magnitude and the affected area can be identified and characterized.

The second measure used is the Precipitation Severity Index (PSI; Caldas-Alvarez et al., 2022b). It identifies extreme precipitation events based on three characteristics: grid point intensity, the affected area, and persistence. First, gridded daily precipitation totals from HYRAS-DE were divided by the climatological 80 % percentile of the corresponding grid point. For
grid points exceeding a ratio of one, these ratios were spatially aggregated and normalized by the affected area. Finally, a temporal summation was applied to grid points exceeding a ratio of one for up to two days prior to the considered event day. The PSI has been successfully implemented in a recent study about heavy precipitation (Piper et al., 2016; Caldas-Alvarez et al., 2022b). Due to the temporal summation, the PSI tends to shift the day of maximum severity by one day compared to HPE$_{\text{crit}}$. Furthermore, PSI detects much more events due to the less restrictive thresholds used.

## 285  3  The July 2021 event in the historical context

As demonstrated in PART1, the disastrous nature of the July 2021 event originated from interactions of atmospheric, hydrological, and hydro-morphological processes and mechanisms that have interacted "optimally" at different spatial and temporal scales. In this section, we put these aspects of the event in a historical and statistical context to evaluate its rarity and to elaborate on probabilities of occurrence.




## 3.1 Meteorological perspective

For the classification of the July 2021 precipitation event in the historical context, we used (i) different thresholds and indices to identify similar historical events from observational data sets, and (ii) the 12 500 years of the LAERTES-EU data to derive more comprehensive precipitation statistics under present-day climate conditions.

### 3.1.1 Event-based analyses

Applying the $\mathrm{HPE_{crit}}$ (Sect. 2.3.2) to the HYRAS-DE data set, in total 26 heavy precipitation events between 1951 and 2021 in Germany turned out to fulfill the criterion (see Table S2 in the supplementary material). The rainfall distribution of the eight most intense events are illustrated in Fig. 2, the events ranked 9th to 20th are displayed in Fig. S1 (see supplementary material). Please note that the actual total event area and precipitation might not be captured by HYRAS-DE in total when the event extended outside Germany. Such cases can not be captured by the presented analysis.

It is striking that almost all events took place during the warm season between May and September and that 11 of these 26 events occurred in the last 20 years with three events in 2002 and three in 2021 alone (Table S2). For the nine top-ranked events, the affected area (within Germany) covers between 6500 and 21 000 km$^2$, while the spatial coverage of the lower-ranked events is below 4000 km$^2$. Furthermore, the majority of events (20 out of 26; 77 %) affected the eastern and southeastern parts of Germany (Figs. 2 and S1). This weather pattern can be related to the typical *Vb* cyclone pathway (van Bebber, 1891) where cyclones move from the Gulf of Genoa in the western Mediterranean along the eastern foothills of the Alps towards central and eastern Europe. Mudelsee et al. (2004) and Messmer et al. (2015), for example, emphasized the role of the cyclone pathway *Vb* and the associated moisture transport on central European river floods during summer, especially in the eastern river basins of Elbe and Oder. The areas with high precipitation totals are usually located on the northern or western flank of these low-pressure areas. Hence, the position of the driving low-pressure system is crucial for the precipitation event reaching into the study area (namely Germany). This is the case for precipitation events close to the border such as the 8 August 1978 (ranked 1st) or the 12 August 2002 (ranked 3rd). Especially in the latter, significant precipitation totals were registered outside Germany (namely in the Czech Republic; e.g., Ulbrich et al., 2003).

The top-ranked event according to $\mathrm{HPE_{crit}}$ is the 8 August 1978, when heavy precipitation of more than 50 mm in 24 hours was registered over almost entire eastern Germany ($\approx 21\,000$ km$^2$; Fig. 2a) leading to a flood event in the Oder river basin (Marx, 1980). The second-ranked event is the 17 July 2002 with an affected area of roughly 13 000 km$^2$ placed over northern Germany. Although highly ranked, no impact information could be found for this particular event which is probably related to the rather flat terrain and thus non-accumulating runoff opportunities in that region. The precipitation event that led to the great Elbe flood in August 2002 (e.g., Ulbrich et al., 2003) ranks 3rd with an affected area of approximately 11 000 km$^2$. This event is also characterized by the highest spatial mean and grid point precipitation. The Berlin event in June 2017 (Caldas-Alvarez et al., 2022a) ranks 4th (approx. 10 000 km$^2$). The July 2021 event is ranked 5th (approx. 8000 km$^2$) in this list (Fig. 2e). Other prominent events are the Elbe and Danube floods from July 1954 (Schröter et al., 2015, e.g.,) with the related precipitation

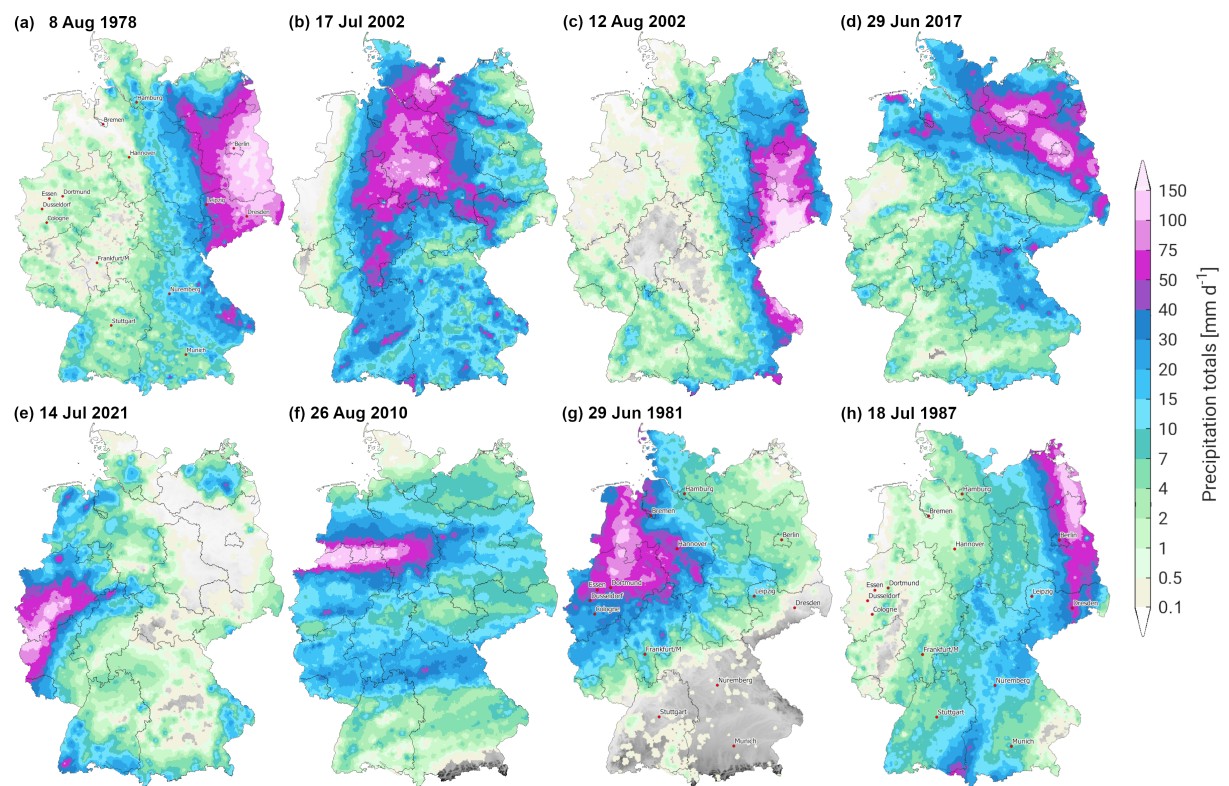

**Figure 2.** The most intense precipitation events in Germany ranked 1st (a) to 8th (h) according to the HPE$_{crit}$ criterion based on daily (05:50 UTC to 05:50 UTC) HYRAS-DE data from 1951 to 2021. The July 2021 event is ranked 5th (e). See also Table S2 for further statistics of the events.

events ranked 9th and 23rd (approx. $6500\,\text{km}^2$ and $1000\,\text{km}^2$) and the flood events in the Alpine region and along the river Rhine in May 1999 (e.g., Frei et al., 2000), which is ranked 15th (approx. $1700\,\text{km}^2$).

Looking at the spatial distribution of the most intense events reveals two main types of precipitation fields (Figs. 2 and S1).
For the first, the daily totals show widespread values above 50 mm embedded in an even larger precipitation field with low to moderate intensities as shown in Fig. 2a–d, g, and Fig. S1a–h. The second type shows a rather small band of very high precipitation intensities above $75\,\text{mm}\,\text{d}^{-1}$ also embedded in different-sized fields of low to moderate intensities. Examples for this type are shown in Fig. 2e, f, and h, and Fig. S1i–l. The July 2021 event (Fig. S1e) shows characteristics of the latter. Note that both types may contain both synoptic-scale precipitation and embedded convection.

For the same list of events, the PSI (Sect. 2.3.2) and API (supplementary material Sect. A) were additionally calculated in order to better capture the preconditions and persistence of the events and their possible role on the impacts. Furthermore, the events were ranked according to their PSI values to compare the results to the HPE$_{crit}$ analysis. While there are only minor changes in the order of the top events, the ranking is more diverse for the less-extreme events (see Table S2). The July 2021
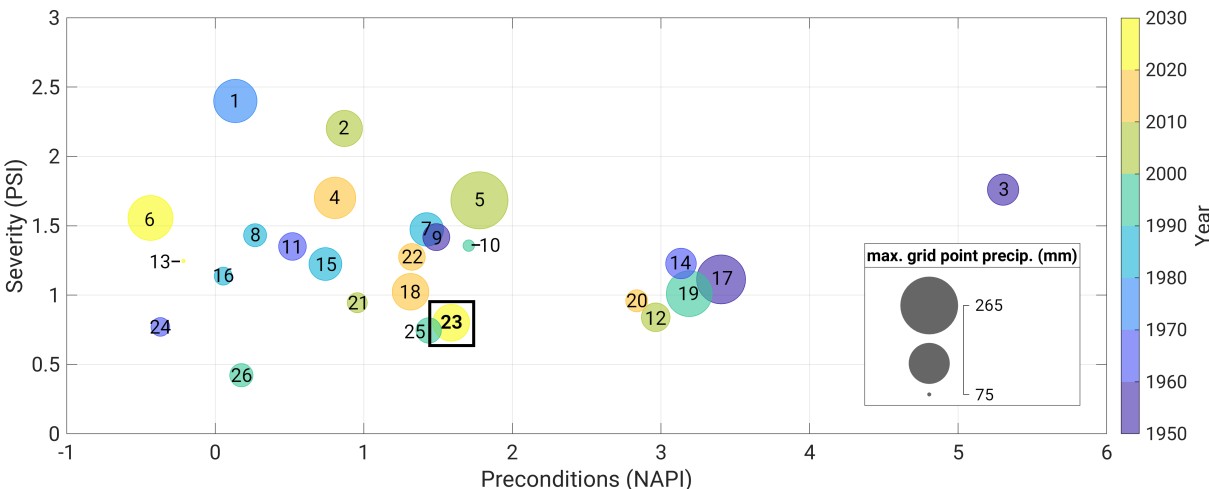

**Figure 3.** Bubble plot of the most extreme precipitation events in Germany according to the HPE$_{crit}$ based on daily HYRAS-DE data (1951 to 2021). The severity (PSI) is given on the y-axis; the x-axis represents the preconditions using the Normalized Antecedent Precipitation Index (NAPI, see supplementary material). The size of the bubbles represents the maximum daily grid point precipitation (small inset legend at the bottom right), while the color indicates the decade of occurrence. The numbers give the ranking according to the PSI analysis (see Table S2). The July 2021 event (ranked 23rd) is marked with a rectangle.

event even ranks only 23rd in the PSI analysis, which is mainly a result of the temporal summation (absence of persistence)
applied in the PSI calculation.

Figure 3 shows the maximum intensities (bubble size) in relation to the soil moisture prior to the initiation of the events expressed by the Normalized Antecedent Precipitation Index (NAPI; see supplementary material Sect. A). NAPI was calculated on and averaged over the same grid pints as PSI. There is a cluster of events at dry to normal or slightly wet preconditions (NAPI between 0.0 and 2.0) and medium severity (PSI between 0.5 and 1.5). The July 2021 event (PSI rank 23) classifies

into this cluster in the less severe but slightly too wet corner. A second cluster appears for wetter preconditions (NAPI around 3.0) and medium severity. Note that the isolated event of 9 July 1954 (PSI rank 3, HPE$_{crit}$ rank 23) results from a persistent precipitation regime that brought already heavy precipitation on 8 July 1954 (HPE$_{crit}$ rank 9) leading to very high NAPI values.

The aforementioned results of the precipitation fields and PSI analyses pointed out a higher level of diversity among the top events in Germany regarding intensity, affected area, and duration. To investigate the relation of the former two in more detail,

the same set of top events is analyzed determining the spatial mean daily precipitation totals on different area dimensions based on HYRAS-DE (Fig. 4). Therefore, we continuously increased the precipitation threshold from 0 mm to event maximum and determined the area of contiguous grid cells that exceeded this threshold. The majority of events accumulate in a band ranging from 95 to 130 mm d$^{-1}$ on an area of 100 km$^2$ and 75 to 110 mm d$^{-1}$ on 1000 km$^2$ to a range of 40 to 80 mm d$^{-1}$ on 10 000 km$^2$ (Fig. 4, gray-shaded area). The top eight events are mostly located above this band (Fig. 4, colored lines). The top-

ranked event (according to the HPE$_{crit}$) of 8 August 1978, for example, is placed well above this band up to area dimensions of

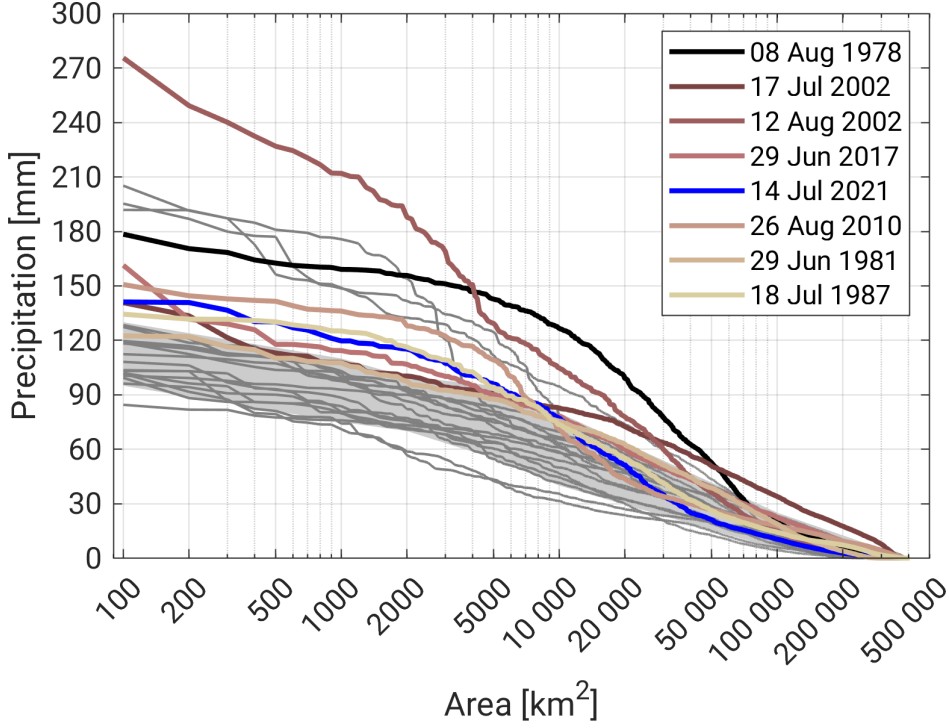

**Figure 4.** Relationship between precipitation totals and affected area (contiguous grid cells) for the 26 strongest precipitation events in Germany based on HYRAS-DE (1951 to 2021) applying the HPE$_{crit}$ criterion (see Table S2). The colored lines represent the top eight events (same as in Fig. 2) with the blue line indicating the July 2021 event. The shaded area marks the range in which more than 50 % of the events are located.

50 000 km$^2$ (Fig. 4, black line). For smaller areas below 4000 km$^2$, the event of 12 August 2002 (HPE$_{crit}$ rank 3) is outstanding. The July 2021 event (Fig. 4, blue line) is also located above the majority band for areas up to 10 000 km$^2$ and within this band for extensions above. For areas above approx. 30 000 km$^2$, the July 2021 event is even placed at the lower boundary of the majority band. Figure 4 underpins the previously shown results in the sense that the July 2021 event was special but not exceptional on small to medium spatial scales regarding the precipitation intensities compared to other historical events.

### 3.1.2 Statistical analyses

In order to derive more comprehensive statistics of precipitation events, the 12 500 years of bias-corrected LAERTES-EU data were used considering the total area CReg. In doing so, the spatial representativeness of the derived statistics is increased and the influence of events occurring randomly at a specific local position is limited. The observed values of the July 2021 event were taken from RADOLAN, which in this context was interpolated to the 0.22° grid of LAERTES-EU for better comparison.

The relation between precipitation intensity and the affected area for 24-hour precipitation totals and different return intervals within LAERTES-EU is shown in Fig. 5. Similar to Fig. 4, we continuously increased the precipitation threshold and
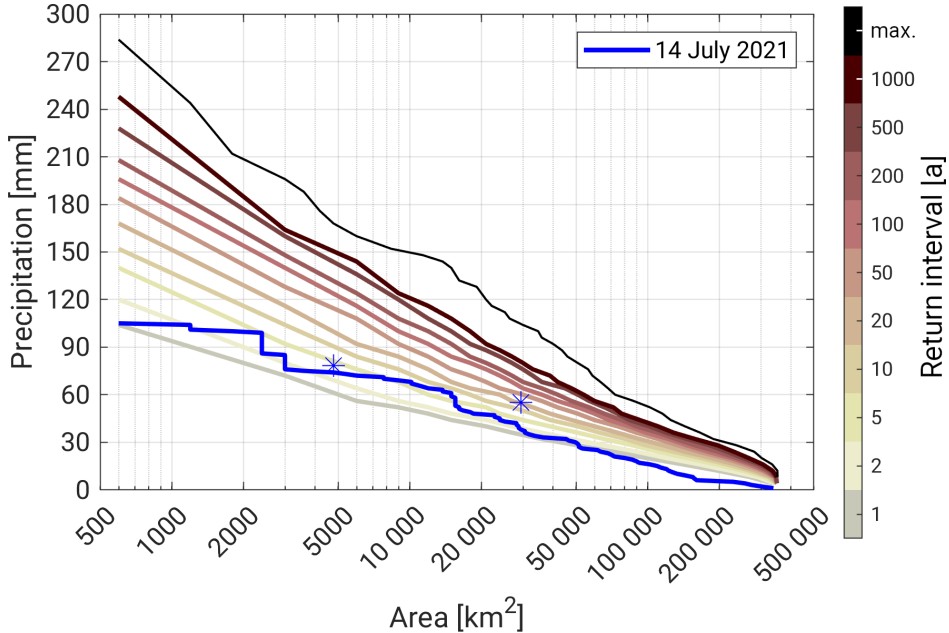

**Figure 5.** Empirical return intervals (colored curves) estimated from the bias-corrected LAERTES-EU data set for precipitation clusters (affected area of contiguous grid points, x-axis) above certain precipitation thresholds (y-axis) for 24-hour totals. The uppermost solid black line represents the maximum values within LAERTES-EU equivalent to a return interval of one in 12 500 years. The blue curve represents the July 2021 event (based on RADOLAN interpolated to the $0.22°$ grid); the two blue marks indicate the spatial means for LReg and SReg.

determined the area of contiguous grid cells exceeding this threshold. In a second step, the number of occurrences of each area–intensity–combination within the 12 500 years of LAERTES-EU is counted which then can be converted to return intervals.

365 The maximum return interval of roughly 10 years was reached for a contiguous area of approx. 15 000 km² with all grid points having 60 mm or more (Fig. 5, blue line). Most of the other size ranges are below the 2-year return level. Using the fixed-sized SReg and LReg areas (Fig. 5, blue marks), the spatial mean precipitation according to RADOLAN during the July 2021 event was 78.4 mm in 24 hours for SReg corresponding to a return interval of 5 years according to LAERTES-EU. For the larger LReg, the observed spatial mean of 55.2 mm corresponds to a return interval of 20 years. Considering only LAERTES-EU data

370 for LReg or SReg, the return intervals are between 100 and 200 years (not shown) which confirms the result of the previous section that most of the events took place in eastern and southern Germany (Sect. 3.1.1). However, comparing the results to the findings in PART1 or Kreienkamp et al. (2021), who both estimated return periods for the event of several hundred to 1000 years based on observations, clearly highlights the uncertainty of a return period estimation solely based on such (short-termed) data sets or for specific areas. Although being exceptional for that particular region, comparable precipitation events

375 occur more frequently in a statistical sense anywhere within CReg.

The results of the spatial analysis are in line with those of PART1 and the findings of the previous section that the extent of the July 2021 precipitation field was special. Furthermore, PART1 illustrated that some members of the DWD weather forecast





ensemble ICON-D2-EPS predicted even higher precipitation totals in the LReg region (cf. PART1, Fig. 4). The maximum
predicted spatial mean precipitation amount for LReg was 78.4 mm in 24 hours, which has an equivalent return interval even
in LAERTES-EU of 500 to 1000 years when considering the total CReg area (3000 years when considering LReg data only)
indicating the hazardous potential of the atmospheric conditions.

In order to also classify and contextualize the above-mentioned results in the context of the 2021 event, the 26 top-ranked
historical events analyzed in the previous section were also put into the statistical LAERTES-EU context deriving the maxi-
mum precipitation return interval and the affected area by this return level (see Table S2, last two columns). The top events
show diverse characteristics with return intervals between less than 1 year and more than 1000 years for areas between 500
and 80 000 km$^2$. Compared to the July 2021 event, there were historical events, for which a lower maximum return interval
was reached at larger affected areas and vice versa. However, only two other events (8 August 1978, HPE$_{crit}$ ranked 1st; and
17 July 2002, HPE$_{crit}$ ranked 8th) reached the same order of return interval (50 or more years) over a comparable order of area
size (10 000 to 20 000 km$^2$), emphasizing that this combination of occurrence probability and extent was exceptional during
the July 2021 event.

### 3.2 Hydrological perspective

In this section, different discharge gauging data sets are used to first classify the July 2021 flood event in a statistical context
in the greater CReg region. In a second step, historical records for the mainly affected Ahr Valley are used to specify the flood
in this particular region.

### 3.2.1 Comparison to GRDC data

In Figure 6, the peak streamflow values (mean daily value) of the 2021GD (in red) and the GRDC data (in blue) are shown
as a function of basin size (up to 1000 km$^2$). For both data sets, a strong and approximately linear dependency between peak
streamflow and basin size is visible. The 2021GD clearly appear at the upper envelope of the GRDC data only exceeded by
five peak values from the GRDC data set. All peak values come from gauges located in mountainous regions with steep terrain
and above-average rainfall, favoring unusually high flood peaks. Two of these peak values were recorded in the Ore mountains
at gauge Dohna (Müglitz) and gauge Pockau (Floha) during the disastrous flood in August 2002 (e.g., Ulbrich et al., 2003),
which is among the top rainfall events shown in Fig. 2 (c). The other three peak values were recorded at gauges in the Black
Forest in southwestern Germany for floods in 1919 at gauge Schwaibach (Kinzig), 1947 at gauge Bad Rothenfels (Murg), and
1991 at gauge Gutach (Elz). Overall, this underlines the exceptional nature of the 2021 flood event, especially when bearing in
mind that the total number of observations in the GRDC data sets covers almost 10 000 years of observations.

### 3.2.2 Comparison to LUBW data

In this section, we put the 2021GD set into the perspective of floods with given return periods. To do so, we first calculated for
each 2021GD gauge the ratio between the 2021 peak flow and the gauge-specific HQ$_{100}$ value (cf. also PART1, Table 1). We

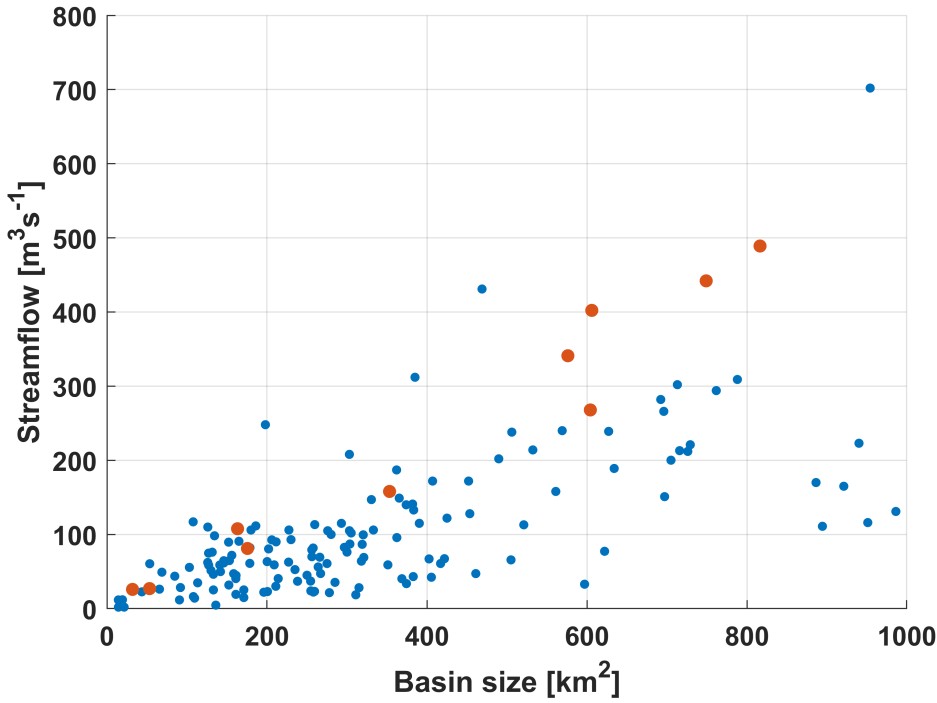

**Figure 6.** Relation between basin size and streamflow for the maximum recorded mean daily values from 124 gauges in the CReg region (GRDC data set; blue dots) and the estimated maximum values of the 2021 flood event at ten gauges in the LReg area (red dots). The location of the latter is shown in PART1, Figure 1, and additional information is given in PART1, Table 1.

used the $HQ_{100}$ value as a reference as it is (i) a widely used design value (e.g., LUBW, 2005), and (ii) as it should be a relatively
robust statistical measure given the on average 51 years of observation at the gauges. The peak factors are shown in Table 1. Please note that we use the estimate of 900 to 1000 $m^3 s^{-1}$ provided by the water administration of RP (cf. PART1, Sect. 3.2) rather than the estimate of 1000 to 1200 $m^3 s^{-1}$ based on hydraulic considerations by Roggenkamp and Herget (2022) for the peak discharge and peak factor calculation at gauge Altenahr (Ahr).

At all 2021GD gauges, the 2021 flood clearly exceeded the $HQ_{100}$, indicated by all peak factors exceeding the value of one,
ranging from 1.7 at gauge Jünkerath (river Kyll) to 7.3 at gauge Bliesheim (Erft) (Table 1). To put these factors into a larger statistical perspective, we calculated a similar peak factor for all gauges in the LUBW data set, but this time between the $HQ_{100}$ and all other return periods. In Table 2, for each return period both the mean and the maximum statistical peak factors from all 355 gauges are shown. The values in Table 2 thus provide a robust reference to broadly classify the 2021GD in terms of return periods.

Even the smallest peak factor of 1.7 from Table 1 places the 2021 flood into the order of magnitude of a $HQ_{5000}$ to $HQ_{10\,000}$ compared to the mean, and $HQ_{500}$ to $HQ_{1000}$ compared to the maximum. The average peak factor of 3.3 (mean of all values in the PF column of Table 1) places the 2021 flood well beyond an $HQ_{10\,000}$, both for mean and maximum. Similar to the




**Table 1.** Gauges of the 2021GD data set in the LReg area (adopted from PART1, Table 1). $HQ_{100}$ indicates a flood with a statistical 100-year return period, $Q_{max,2021}$ is the peak discharge of the July 2021 flood event (values are estimates), The peak factor PF is defined as $Q_{max,2021}$ divided by $HQ_{100}$.

| Gauge/river name | $HQ_{100}$ ($m^3\,s^{-1}$) | $Q_{max,2021}$ ($m^3\,s^{-1}$) | PF |
|---|---|---|---|
| Müsch/Ahr | 152.0 | ≈ 500 | ≈ 3.3 |
| Altenahr/Ahr | 241.0 | ≈ 1000 | ≈ 4.0 |
| Jünkerath/Kyll | 118.0 | ≈ 200 | ≈ 1.7 |
| Kordel/Kyll | 248.0 | ≈ 600 | ≈ 2.5 |
| Prüm 2/Prüm | 51.6 | ≈ 120 | ≈ 2.3 |
| Prümzurlay/Prüm | 278.0 | ≈ 600 | ≈ 2.0 |
| Schönau/Erft | 19.0 | ≈ 100 | ≈ 5.2 |
| Bliesheim/Erft | 71.0 | ≈ 500 | ≈ 7.3 |
| Hückeswagen/Wupper | 73.0 | ≈ 200 | ≈ 2.9 |
| Opladen/Wupper | 250.0 | ≈ 530 | ≈ 2.1 |

**Table 2.** Statistical peak factors PF for the LUBW data set. The statistical peak factor is defined as $HQ_X$ (X = 200, 500, 1000, 5000, 10 000) divided by $HQ_{100}$, separately for each gauge in the data set. $PF_{mean}$ is the mean peak factor of all 355 gauges, and $PF_{max}$ is the largest factor.

| | $PF_{mean}$ | $PF_{max}$ |
|---|---|---|
| $HQ_{200}$ | 1.13 | 1.22 |
| $HQ_{500}$ | 1.32 | 1.53 |
| $HQ_{1000}$ | 1.47 | 1.83 |
| $HQ_{5000}$ | 1.63 | 2.23 |
| $HQ_{10\,000}$ | 1.86 | 2.84 |

comparison with the GRDC data, this underlines the exceptional nature of the 2021 flood. Finally, the maximum peak factor of 7.3 from Table 1 is so far beyond the peak factors in Table 2, and hence, so far beyond an $HQ_{10\,000}$ that it raises doubts about the

validity of evaluating the 2021 flood in terms of statistical return periods using short-run flow data. We will further investigate this matter in the next subsection.

### 3.2.3 Comparison to historical data

In this section, we focus on gauge Altenahr (river Ahr), which is selected for two reasons: first, it is placed in a region that is among the most severely affected by the 2021 flood (cf. PART1); second, long-term historical records are available for this

gauge, which are helpful to illuminate the problem of estimating return periods of exceptional flood events.

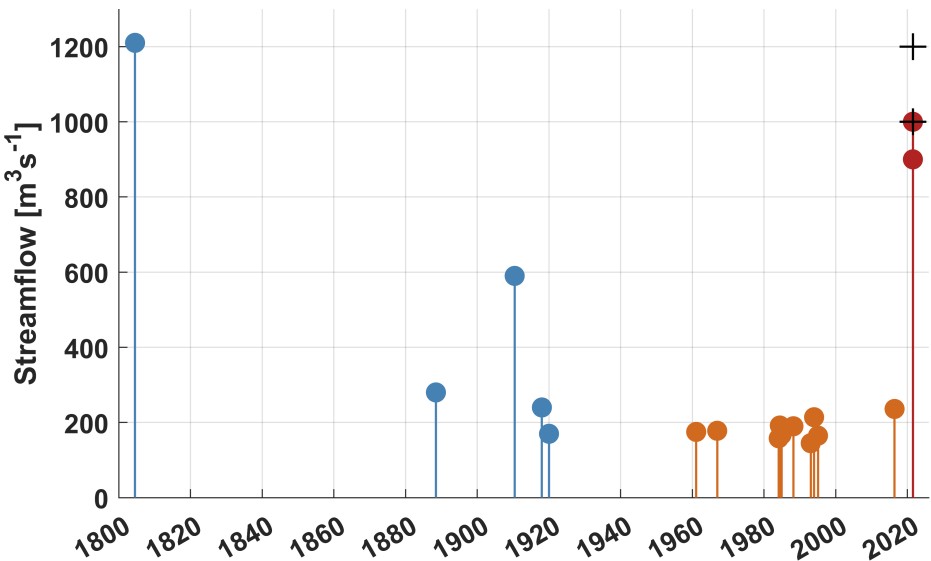

**Figure 7.** Time series of recorded and reconstructed flood peaks at gauge Altenahr (Ahr) (modified from Roggenkamp and Herget, 2022). The blue dots are reconstructions, the orange dots are gauge recordings (starting 1946); the red dots indicate the range of the 2021 flood peak estimated by the water authority of Rhineland-Palatinate; the black crosses indicate the range of the 2021 flood peak estimated by Roggenkamp and Herget (2022) based on gauge recordings and reconstructions.

Figure 7 shows the available flood events at gauge Altenahr. The events in orange are from gauge recordings starting in 1946 and were the basis for the HQ$_{100}$ estimate (241 m$^3$ s$^{-1}$) by the water administration of RP. The blue dots are historical floods, notably the 1804 and 1910 events, for which peak discharge estimates are provided by Roggenkamp and Herget (2014a), Roggenkamp and Herget (2014b), and Roggenkamp and Herget (2022). The red dots indicate the range of the 2021 flood

peak estimated by the water authority of RP and the black crosses indicate the range of the 2021 flood peak estimated by Roggenkamp and Herget (2022) based on gauge recordings and reconstructions. It is clearly visible that the gauge recordings since 1946 missed several major historical flood events, which renders the HQ$_{100}$ based only on these values non-representative. Acknowledging this, the water authority of RP recently provided an updated HQ$_{100}$ estimate of 434 m$^3$ s$^{-1}$ based on the entire time series from 1804 to 2021 as shown in Fig. 7 (Hennrichs, 2022). Based on the updated HQ$_{100}$, the peak factor for gauge

Altenahr of the 2021 event reduces from approx. 4.0 (see Table 1) to approx. 2.3. Comparing the updated value to the values in Table 2 still places the 2021 event at Altenahr in the order of magnitude of an HQ$_{10\,000}$ compared to the mean, and an HQ$_{5000}$ to HQ$_{10\,000}$ compared to the maximum.

Vorogushyn et al. (2022, in review) also estimated the return period of the 2021 flood at gauge Altenahr. While their estimated return period of the 2021 flood based on recorded data from 1949 to 2019 is larger than 10$^8$ years, which is very unrealistic

and clearly shows the limitations of extreme value statistics for rare events based on non-representative samples, the same authors then estimated the return period of the 2021 flood to be in the order of magnitude of HQ$_{10\,000}$ and an HQ$_{100}$ of approx. 300 m$^3$ s$^{-1}$ when adding historical floods since 1804. Comparing the corresponding peak factor of approx. 3.0 with the values





in Table 2 also places the event roughly into the order of magnitude of $HQ_{10\,000}$. It is noteworthy that the authors report difficulties when fitting the GEV distribution to the data and suggest the existence of a mixed rather than single distribution, where the extreme floods of 1804, 1910, and 2021 come from a separate distribution. A possible explanation for such a mixed flood distribution could be the existence of rare but extreme weather situations responsible for unusually large floods, and/or the onset of special rainfall-runoff mechanisms only in the case of extreme rainfall. Once more, this underlines the challenges of extreme value statistics and the large uncertainties when estimating return periods for the 2021 event. It also indicates the need for even longer historical time series and reconstructions as far as possible and/or the examination of the completeness of the events between 1804 and 1900 as well as before 1804, where over 70 floods occurred in the Ahr River basin since the year 1500, including the large 1601 event (Seel, 1983). In addition, 1818 and 1848 were also large events with currently no reconstructed streamflows.

### 3.3 Hydro-morphodynamical perspective

In this section, we analyze hydrodynamic and morphodynamic processes which occurred in both, the Ahr and the Erft Valley, with the main focus on (i) the anthropogenic influence on the processes which evolved dramatically in the last century, (ii) elements such as debris and sediments which enhance the flood hazard, and (iii) the landscape organization and occupation which conditions the downstream hazard.

The inclusion of sediment and linked geomorphic processes in the analysis of flood risk is discussed by several authors (e.g., Best et al., 2022). However, in the images taken in the aftermath of the July 2021 floods in Germany, it is striking to see that sediments contributed only a small portion to the total accumulated debris, which invaded the river network, streets, rural spaces, culverts, and buildings. There is an obvious and significant difference from similar images made one hundred or more years ago. The nature of the debris, which accumulates in narrower cross-sections in the urban space (streets) and river network (bridges), changed considerably. There is clearly an anthropogenic influence contributing to new types of debris of industrial origin. Besides sediments and the overwhelmingly present natural dead wood and vegetation, we observed a high volume of industrial elements such as vehicles and caravans, bins and containers, and construction materials, which exacerbated the hazard in the river systems.

We observed that parts of the Ahr Valley which historically would be occupied during flood events, and which often are preferential areas for deposition of sediments and debris transported by the flow, are now urban settlements. Such is the case of the southern part of the urban settlement of Schuld, which developed in the inner region of an Ahr Valley bend. This flow region is prone to sedimentation and a natural landscape sink of sediment and debris, which is transported by the river during floods. Parts of the town of Altenburg (Altenahr), bordering the right bank of the river Ahr, were constructed on an abandoned oxbow lake. The paleogeography of the valley shows that this region was an old meander of the river Ahr. Although dry under normal flow conditions, the abandoned oxbow lake is of alluvial nature and prone to inundation in high flow situations. Furthermore, being a low-velocity flow region, it is prone to sedimentation and settling of debris transported by the flood (Dépret et al., 2017). To a very limited extent, the inundation of this area may have contributed to the flow lamination and attenuation of the


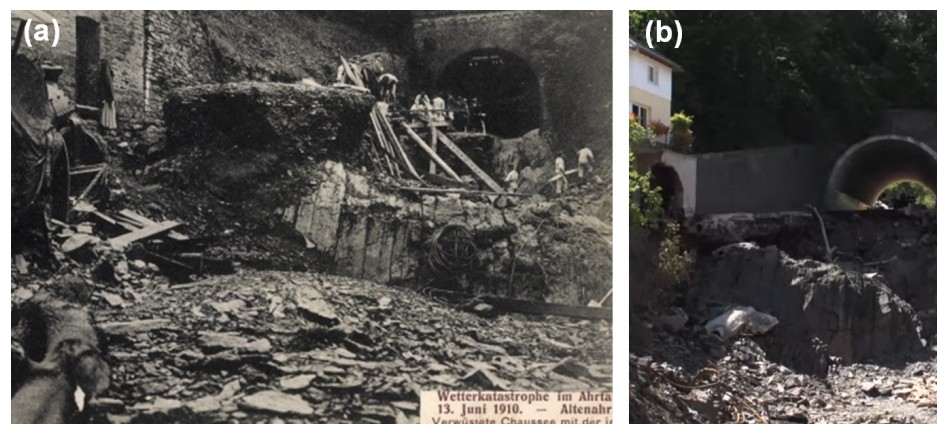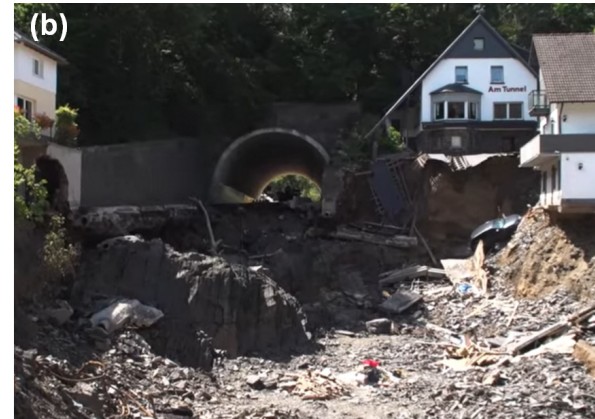

**Figure 8.** Downstream end of the road tunnel in Altenahr after (a) the flood of 1910 (photo credits: *Kreisarchiv Ahrweiler*), and (b) after the flood of 2021 (photo credits: Dieter Könnes).

downstream peak discharge. Nevertheless, the main consequences of the inundation of this oxbow lake are negative, namely the destruction of the urban area settled herein as described in PART 1.

Another example of the consequences of the historical transformation of the morphology of the valley over time is the case of Dernau. As discussed in PART1, the construction of a valley bottleneck increased substantially the flood levels recorded in July
2021 when compared to a similar flood in terms of peak discharge in 1804. In this particular case, we argue that morphological changes in the valley of anthropogenic origin may be the reason for more severe inundation levels during the 2021 flood event. This transformation was mainly materialized during the construction of the railroad (*Ahrtalbahn*) in the 1880s with several bridges that created some bottlenecks downstream the town of Dernau.

As shown in PART1, the occurrence of a bypass in the landscape due to the road tunnel in Altenahr (Ahr) is an example of
an anthropogenic landscape singularity that disrupted the continuity of the valley for extreme conditions as those in July 2021. In this case, the tunnel in Altenahr bypassed the flood through, which may have probably eliminated the lamination effect of one meander of the river Ahr. To a limited extent, it is expected that the downstream peak discharge was exacerbated. The main effect which is visible, however, must have been provoked by the substantial and sudden reduction of flow in the meander. Since the flood water was charged with sediments under suspension, substantial deposition occurred in the meander with the
occurrence of the bypass changing considerably the channel morphology. The road tunnel in Altenahr was constructed in 1834, and notably, the same flow bypassing happened already in the past during the 1910 dramatic flood in the valley (Fig. 8). The final configuration of the slope and tunnel toe after the passage of both floods show striking similarities. An uncomfortable sensation of *déjà-vu* suggests the importance of considering historical records and information in the preparation and planning for future flood events.

Besides bridges or tunnels, which are the most visible and paradigmatic historical changes of anthropogenic nature in the valley morphology, the construction of buildings, infrastructures, and industrial equipment add singularities to the landscape, which are capable of dramatic landscape alteration. Such was the case of the large-scale erosion episode that originated in





the mining pit in Blessem (Erftstadt) in the Erft catchment. This mining pit, which was under exploration, showed to be a singularity in the landscape capable of triggering an impressive process of retro-dendritic erosion. The cavity formed due to
this process caused the destruction of houses and endangered furthermore a changing of the course of the river caused by the breach and collapse of the Erft riverbank that endangered the nearby highway. The occurrence of such landscape changes, which have been spread in the territory over the last century needs further assessment. They can introduce local landscape fragility as the one in Blessem, but they can also be used as opportunities to retain flood volumes having a positive effect on floods attenuation, as also happened in Blessem (cf. PART1).

During the July 2021 event, the destruction of gauging stations was particularly visible, for instance, in Altenahr (river Ahr). The destruction of these has negative consequences for flood management prior to and during emergencies since they are essential instrumentation to observe in real-time the evolution of floods and to keep historical hydrometry records of the valley. The gauging stations are destructed by local scour or bank erosion and collapse. Analysis of the historical stability of the position, where they are placed, could inform a safer installation. Furthermore, the use of remote gauging in locations
less susceptible to morphological changes during floods could be considered using videos (e.g., Le Coz et al., 2010), or other remote sensing techniques, to monitor hydrological quantities in the rivers.

The hazardous effects of the meteo-hydrological extreme event herein described, ultimately depend on the landscape organization and occupation, and on the stability of the landscape and river morphology which is variable in time. Until now, framework documents such as the Floods Directive (*Directive 2007/60/EC of the European Parliament and of the Council*
*of 23 October 2007 on the assessment and management of flood risks*) are merely suggestive in the consideration of transported sediments and debris in flood risk assessment. Consequently, the consideration of moving boundaries and time-evolving hydro-morphodynamic processes in flood modeling and emergency planning is not common in Europe. Flood managers often consider a rigid landscape in flood risk assessment. Lane et al. (2007) argued on the need to consider the joint synergistic impacts of the combination of climate change and sediment imbalance in rivers, whereas Nones et al. (2017) explicitly defends
the inclusion consideration of hydro-morphology and sediments in the implementation of the Floods Directive. Lucía et al. (2018), who studied the 2016 flash flood in Braunsbach (Germany), suggested an update of the Floods Directive to require that flood risk assessment includes a heuristic hydro-morphodynamic approach, which considers the landscape and river channels network.

## 4  Relation to climate change

In this chapter, we examine the 2021 flood in the context of future climate change. With this aim, it is of particular interest how a specific extreme event such as the July 2021 event would unfold under different climate conditions, and more generally, what changes regarding the intensity and frequency of extreme precipitation events can be expected in the future. For the first, PGW simulations and subsequent hydrological discharge modeling were performed, and for the latter, conventional climate projections at the CPM scale (KIT-KLIWA) were considered in this study.


### 4.1 Pseudo-global-warming experiments

#### 4.1.1 The July 2021 precipitation event in a warming climate

Figure 9 shows the precipitation totals of PGW simulations of the July 2021 event against the background of different temperature perturbations. The control simulation (Fig. 9b) based on unperturbed ($\pm 0.0$ K) ERA5 initial and boundary conditions adequately reproduces the event in its magnitude and location (see also Fig. 1b for comparison). The simulated precipitation totals in LReg ($56.5$ mm d$^{-1}$) are almost equal to the observed totals ($55.4$ mm d$^{-1}$ from RADOLAN; cf PART1). Likewise, the most intense precipitation was simulated over the affected region. For SReg, the simulated precipitation totals ($88.5$ mm d$^{-1}$) are slightly higher in comparison to the observed totals ($75.2$ mm d$^{-1}$ from RADOLAN). This overestimation might be due to a second simulated precipitation peak in the western part of SReg, which is not visible in the observations (see Fig. 1b).

Considering the PGW simulations, precipitation is lower under colder pre-industrial-like climate conditions (e.g., –1 K; Fig. 9d) over LReg with a decrease of 11 % ($50.3$ mm d$^{-1}$), while warmer conditions (e.g., +2 K, corresponding to GWL3, Fig. 9c, d) lead to 18 % higher precipitation totals over LReg ($66.7$ mm d$^{-1}$). Over SReg, the precipitation also decreases by 11 % for a –1 K cooling, and increases by approx. 11 % for a +2 K warming. Taking into account all PGW simulations (Fig. 9d, solid lines), the relationship between temperature and precipitation change follows indeed the Clausius-Clapeyron (CC) scaling (7 % per 1 K) for the smaller SReg and even a super-CC scaling (9 % per 1 K) for the larger LReg. This is in line with recent findings about the relationship between temperature and precipitation, which found that daily precipitation extremes mostly increase at approximately the CC-rate (e.g., O'Gorman, 2015; Trenberth et al., 2003).

By considering return periods based on the LAERTES-EU data (see Sect. 3.1.2) and the PGW precipitation outputs, the change in the intensity of such an event under global warming can be assessed. For SReg the return period of the control run is about 10 years. For –1 K, the return period is reduced to only 5 years, while for the +2 K simulation a return period of about 20 years is estimated. For LReg, the differences in return periods became even higher. While for the control run, a return period of 20 years can be assigned, the return period for the +2 K simulation increased to 200 years, emphasizing the hazardous potential of such an event in the context of future climate warming. This becomes even more evident when considering LAERTES-EU data for LReg (SReg) only (not shown). The return periods of the control run for these specific areas are around 100 to 150 years for LReg and 400 to 500 years for SReg. At –1 K, the return periods reduce to 50 to 60 years for LReg and 150 to 200 years for SReg, while for +2 K there is an increase to approx. 500 years for LReg and approx. 800 years for SReg. These are statistically rather robust estimates due to the total length of LAERTES-EU of 12 500 years.

#### 4.1.2 Hydrological response

The first test with LARSIM-Ahr using the original PGW precipitation fields revealed a mismatch and non-conclusive results for the resulting peak flows (not shown) due to slight spatial shifts of the rainfall centers between the scenarios. The resulting rainfall totals in the rather small ($749$ km$^2$) Ahr river basin did not reflect the overall areal rainfall decrease or increase of the PGW scenarios. To overcome this shortcoming, a more robust approach is applied determining correction factors for the two considered PGW scenarios –1 K (PGWcold) and +2 K (PGWwarm) compared to the PGW control simulation (PGWcontrol,


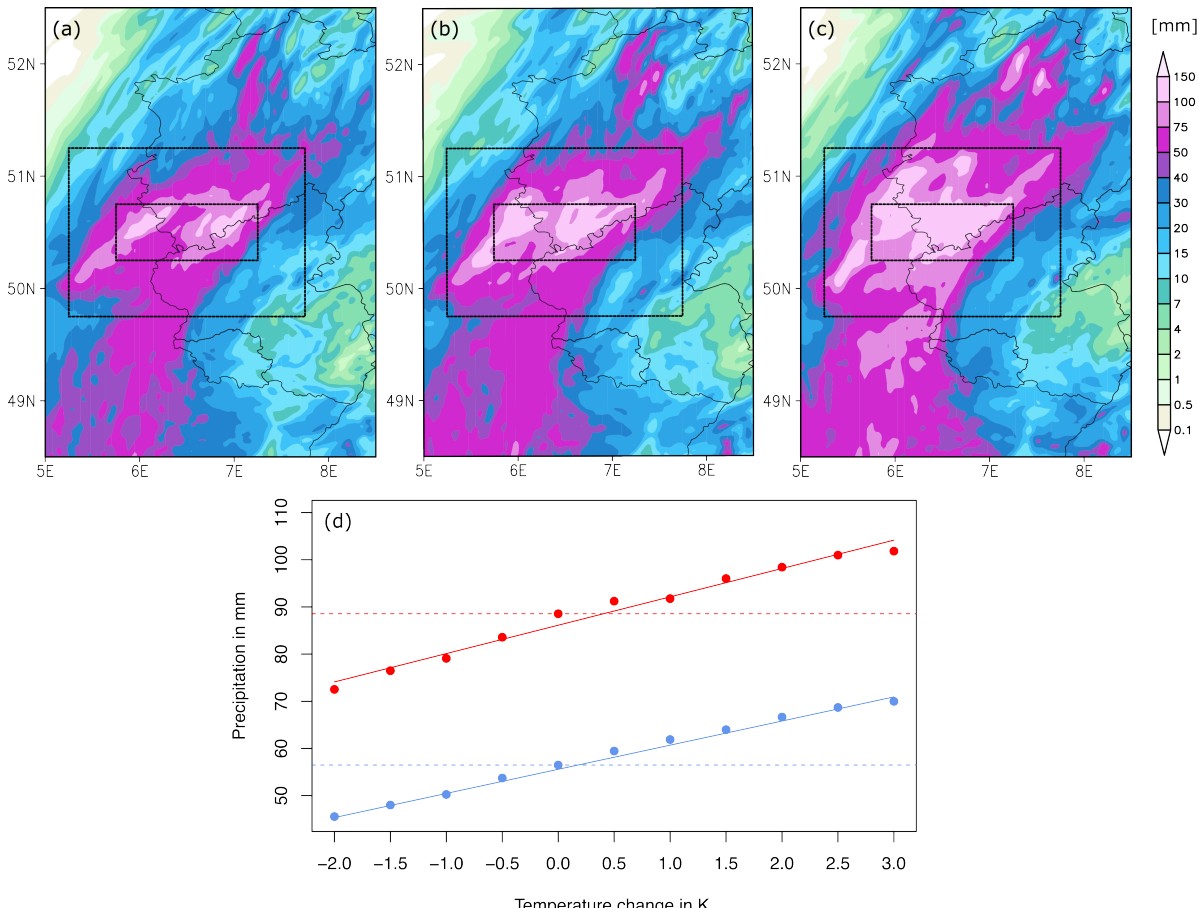

**Figure 9.** 24-hour precipitation sums (14 July 2021, 06:00 UTC to 15 July 2021, 06:00 UTC) from PGW experiments using the WRF model. Horizontal distributions for (a) PGW −1 K, (b) control run ±0 K, and (c) PGW +2 K. (d) Area averaged 24-hour precipitation sums (LReg in blue, SReg in red) plotted against temperature change for all conducted PGW experiments. Solid lines represent linear regression lines that are used for calculating the correction factors for the hydrological discharge modeling (see Sect. 4.1.2). Stippled horizontal lines denote the mean 24-hour precipitation amount of the control run (56.5 mm for LReg, and 88.6 mm for SReg).

±0 K). As demonstrated in the previous section, the spatially averaged rainfall totals for LReg of PGWcontrol match the observed totals from RADOLAN. Using the derived linear regression function for LReg (Fig. 9d, solid blue line), the ratios

between the PGWcontrol and the scenarios PGWcold (PGWwarm) were calculated. The resulting adjustment factors were 0.92 for PGWcold, and 1.16 for PGWwarm representing the climate change signal compared to present-day conditions. The observed rainfall data set used for the LARSIM-Ahr reference simulation was then multiplied by these factors and used as input for two additional LARSIM-Ahr scenario simulations.

Figure 10 shows observed and simulated streamflow time series at gauge Altenahr (Ahr). The black line shows the values

based on observation and reconstruction (cf. PART1 for details), the peak value of 991 $m^3\,s^{-1}$ corresponds to the upper red


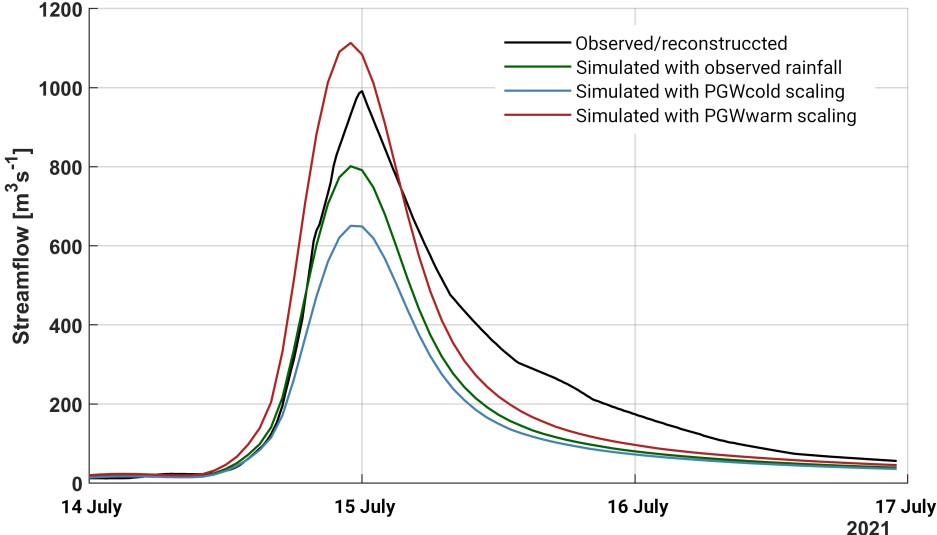

**Figure 10.** Streamflow at gauge Altenahr (Ahr). Black is from observation and reconstruction (cf. PART1), green is from LARSIM-Ahr simulation forced with observed rainfall (Bardossy et al., 2022), blue is from simulation with PGWcold rainfall scaling, and red is from simulation with PGWwarm rainfall scaling.

dot in Figure 7. The green line shows the simulated streamflow time series using LARSIM Ahr with the best available rainfall observation product as input. The corresponding peak flow of $801 \, \mathrm{m^3 \, s^{-1}}$ underestimates the observed one by about 20 %. This can be attributed to deficiencies in the hydrological model, the initial conditions, and the used rainfall product. Nevertheless, the overall magnitude and course of the event are well captured. The blue and red lines show the results of the streamflow simulations based on the observed rainfall scaled by the ratios of PGWcontrol to PGWcold and PGWcontrol to PGWwarm, respectively. The resulting simulated peak flows of $650 \, \mathrm{m^3 \, s^{-1}}$ (PGWcold) and $1113 \, \mathrm{m^3 \, s^{-1}}$ (PGWwarm) are smaller (larger) than the reference by a factor of 0.81 (1.39). For both the cooling and the warming scenario, the hydrological response was, therefore, more pronounced than the meteorological one. For PGWcold, a precipitation decrease of 11 % was simulated while the LARSIM-Ahr simulations show a decrease of the flood peak of 19 %. This non-linear response is even more pronounced for PGWwarm, where a precipitation increase of 18 % for LReg caused a 39 % increase in the flood peak. This indicates that increasing meteorological hazards due to climate change may be amplified by the rainfall-runoff transformation.

## 4.2 High resolution future climate projections

The evolution of heavy precipitation in a warming climate is now investigated in conventional climate projections using the KIT-KLIWA ensemble considering exemplary the 10-year return value (RV10) of spatially averaged daily precipitation. Figure 11 shows the development of RV10 with ongoing global warming for each of the four ensemble members for CReg. The distribution of return values within CReg is represented in the box plots. For the reference period (1971 to 2000; equivalent to GWL of +0.46 K compared to pre-industrial conditions), the RV10 averaged over CReg is 54.0 mm for the ensemble mean





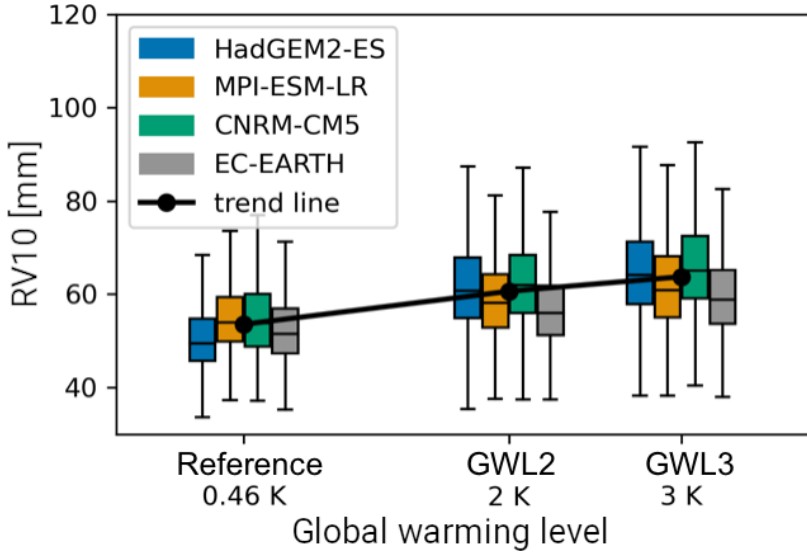

**Figure 11.** Box plots of spatially averaged RV10 distributions over CReg for the reference period, GWL2, and GWL3 for all ensemble members of KIT-KLIWA. The boxes represent the lower and upper quartiles, the center lines the median, and the whiskers outliers defined as more than 1.5 times the interquartile range. Black dots and line show the multi-model ensemble mean for each set of runs and the trend.

(Fig. 11, black dots and line). For comparison, the RV10 in KOSTRA (cf. PART1 and Malitz and Ertel, 2015) averaged over the German part of CReg is 64.0 mm, so the KIT-KLIWA result is about 10 mm (or 18 %) smaller. However, the model-based value

was calculated for fixed calendar days, whereas KOSTRA gives the maximum precipitation in an arbitrary 24-hour interval. The German regulations *DWA-Regelwerk 531*, which KOSTRA is based on, proposes a factor of 1.14 to correct this methodical difference. Applying this technical correction to the KIT-KLIWA data, the adjusted RV10 is 61.6 mm (4 % difference to KOSTRA). Furthermore, Junghänel et al. (2017) specified a uncertainty range of ± 15 % for a 10-year return interval in KOSTRA. Taking both into account, the KIT-KLIWA data are in the range of KOSTRA.

For each ensemble member, the average value increases with proceeding global warming (Fig. 11). The extent of this increase measured as the normalized difference between GWL3 and reference period varies slightly between the simulations with the strongest (lowest) increase in the CNRM-CM5 (EC-EARTH) driven simulation. Averaging over CReg and all ensemble members, a trend of 4 mm per degree of warming is predicted. This corresponds to a relative change of 8.4 %. Considering the LReg only, the increase is about the same magnitude with 7.8 %, and also the individual ensemble members show similar

behavior (not shown).

The analysis of additional return values showed that the magnitude of the relative increase per degree of warming depends on the return period with lower (higher) rates for shorter (longer) return periods (Table S3). For CReg, the increase between the reference period and GWL3 is only 5.6 % for the 1-year return interval, and 7.5 % for the 5-year return interval. For longer return periods (e.g., 30 years), the relative increase is higher with 10.1 % per degree of warming. Thus, the scaling of heavy

precipitation with global warming is projected to be in the range of the Clausius-Clapeyron scaling (7 % per 1 K warming) for


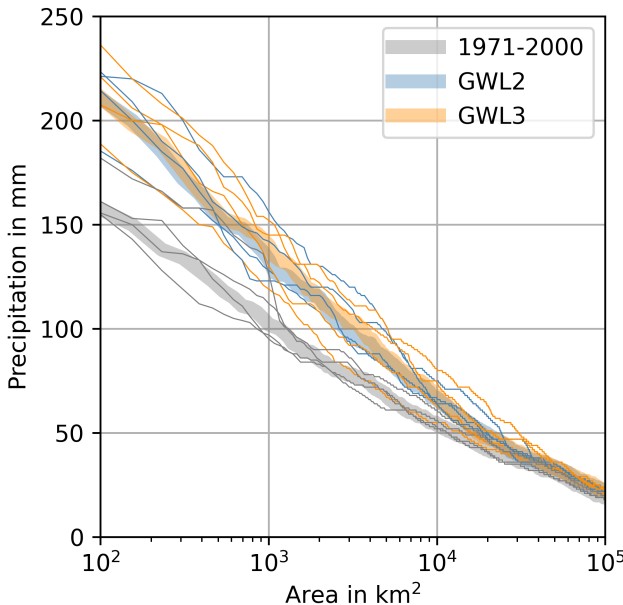

**Figure 12.** Spatial extent of daily precipitation clusters (contiguous grid points) above the RV10 estimated from the KIT-KLIWA data set for the present-day reference period, GWL2, and GWL3. Thin lines represent individual ensemble members, and thick lines the ensemble mean.

return periods of 3 to 10 years. For LReg, the climate change signal is about 0.5 to 1.0 % per degree warming smaller than for CReg for all considered return periods. Analysis showed that this increase is not purely linear, but a slightly stronger increase is observed when considering the change from the reference period to GWL2 only (Table S3). However, the deviation from the linear trend is within the large range of variability of the ensemble.

As already elaborated at the beginning of the paper, the severity of a precipitation event depends not exclusively on the local intensities. The affected area is another essential factor especially for triggering floods. Analogous to the analyses in Sect.3.1.2 based on the LAERTES-EU data set, a cluster analysis was conducted for the KIT-KLIWA simulations to investigate the development of the spatial extent of RV10 in a warmer future.

    The projections show that the extent of precipitation loads experienced today as an RV10 will increase in a warmer climate
(Fig. 12). The analysis of the single ensemble members (thin lines in Fig. 12) shows that all four members as well as the ensemble mean (thick lines in Fig. 12) agree on the increase of larger events in a warmer climate. However, though the difference from the reference period to GWL2 is clearly visible, only a few changes are projected on average from GWL2 to GWL3. The small spread between the ensemble members in the reference period, especially for large areas (more than $1000 \, \text{km}^2$), increases at GWL2 and GWL3. The relative increase is almost constant for a wide range of areal dimensions of contiguous clusters. For
example, a cluster with daily precipitation totals of at least $150 \, \text{mm}$ nowadays occurs at an extent of about $150 \, \text{km}^2$ while such an event is expected to extend about $600 \, \text{km}^2$ for GWL2 and GWL3. The present-day intensity of a cluster of $600 \, \text{km}^2$ is about





115 mm; the future intensity for a cluster of 150 km² is about 200 mm, which in both cases is an increase of about 30 %. For areas larger than 20 000 km² no changes can be resolved by the analysis anymore, which is also influenced by the limited area of the model domain, bounding these large extreme events.

The results presented above indicate that the projected precipitation change for high return values might even exceed the theoretical CC-scaling (super-CC scaling) over the south and central Germany in a warming climate. Comparing the results with the PGW simulations for the July 2021 event (see Sect. 4.1.1), the relative changes within the conventional climate projections of KIT-KLIWA are of similar magnitudes. In comparison with the RV10 in LAERTES-EU (Fig. 5), the RV10 in KIT-KLIWA is smaller by about 30 % over the entire range of analyzed spatial dimensions. This discrepancy might largely be attributed to

the difference in ensemble size (factor 10) and a varying bias correction. Additional uncertainties arise from different methods of estimating the RV10 (GPD in the case of KIT-KLIWA and empirical in the case of LAERTES-EU).

### 4.3   Further considerations on flood management under a changing climate

Harrison et al. (2019) concluded that different landscape settings respond differently to changes in climate. Consequently, the evaluation of the effects of global warming on flood hazards is not restrained to considerations of extreme meteorological

events. The physical processes causing flood hazards are a combination of extreme meteorological events with the landscape occupation and organization and with the stability of the landscape and river morphology. These factors, together with the societal response to emergencies, define the level of flood hazard of a valley.

Climate conditions the landscape occupation and organization, so it is expected that global warming will cause changes in the land cover due to the adaptation of natural elements such as vegetation and soil. Furthermore, it will force the adaptation

of human activities and soil use, for instance by adaptation of agricultural practices. These factors, in turn, have a direct effect on the surface flow, and on the influx of inorganic sediment (Eekhout and de Vente, 2022) and large debris such as deadwood. Rivers are dynamic systems, where non-linear interactions occur between the flowing water, the boundaries, which can be erodible or fixed, moving sediment, and debris and vegetation (Seminara, 2010). Imbalances in river systems, for instance, changes in discharge, riverine aquatic and riparian vegetation, and sediment production can have dramatic effects on the fluvial

morphology stability and flow conveyance (Crosato et al., 2022). These aspects are all vulnerable to climate change.

As seen in the analysis of the July 2021 extreme floods, this dynamic character of valleys and rivers ultimately impacts the flood hazard. The non-linear interactions and morphological changes are a permanence in rivers occurring at these different time scales. Beaty (1974) and Kern (2013) conceptualized a division of flow events in rivers between those corresponding to equilibrium processes and those corresponding to, what they call, catastrophic processes, those that represent dramatic changes

in the river morphology. The second type of event is the one scaring the landscape and river channel networks, changing the boundaries with which static flood modeling is commonly done. Long-term channel bank and slope adjustments correspond to long-enough periods during which one may assume steadiness in channel morphology (several decades) and are easily incorporated by a classical flood risk analysis based on fixed boundaries. Mid-term reach adjustments, happen in a multi-year time scale typical inferior to the return period which is considered on flood risk assessment (see discussion by Bung (2021) based on

the current extreme events), hence requiring at least the analysis of the sensitivity of flood modeling results to morphological





changes in flood risk assessment (Pender et al., 2016; Radice et al., 2016). Short-term local-to-reach adjustments occur almost instantly or on a timescale corresponding to the duration of the flood event (minutes to days), for which the consideration of dynamic changes of river morphology during events is necessary (Dietze and Ozturk, 2021). Considering the frequency of occurrence, Magilligan et al. (1998) argued that the frequency of occurrence of flood events and river morphological adjustments

may not always coincide. The channel-forming flow in a river typically corresponds to a hydrological flood with a return period of 1 to 10 years (Copeland et al., 2000; Doyle et al., 2007; Annable et al., 2011), depending on many other factors such as landscape occupation, slope, valley confinement, stream power flood period and sequence (e.g., Magilligan et al., 1998; Lucía et al., 2018). These two important factors for flood hazard evaluation and implementation of flood mitigation measures, frequency, and duration of extreme events, are susceptible to climate change bringing hence extra complexity for flood managers.

Taking a historical perspective, there are two main aspects that influence the flood hazard in valleys, which are important to highlight: the time-varying character of the valleys, which happens at scales that are not necessarily compatible with the return periods considered for flood management, and the anthropogenic changes in valleys, which, besides being responsible for the urban occupation, have a direct influence in channels morphology and an indirect influence on the type of debris reaching the rivers. These two aspects correspond to a twofold intersection with climate change: the changes that climate scenarios may

have on landscape and river morphodynamics (uncertain weathering and alteration in land cover) are still to be unraveled.

Finally, in a context of an uncertain and changing climate, a probabilistic analysis of the combination of landscape oddities, as some herein described, with extreme meteorological events is necessary to stress test the safety of valleys against improbable and unforeseen occurrences such as the ones occurred in July 2021 in Germany.

## 5  Discussion and conclusions

The present work is the second part of an interdisciplinary study describing, analyzing, and classifying the disastrous July 2021 precipitation and flood event in Central Europe. The main focus of the second part was on (i) the comparison and classification of this event in a historical and statistical context and (ii) the assessment of future climate change effects on such extreme events. The main conclusions with respect to the research questions from Sect. 1 are as follows:

(I) *How does the event classify within the historical context of precipitation and flood events?*

The July 2021 precipitation event was among the most intense historical events in the past 70 years in Germany (ranked 5th). However, the statistical analyses using the LAERTES-EU RCM ensemble revealed an underestimation of return values derived using observational records given their limited length. The observed discharges of the July 2021 flood event along the river Ahr were extreme regarding statistics based on observations, but not extraordinary when additionally considering reconstructed historical events (e.g., 1804, 1910), demonstrating that the existing flood hazard maps at the

time did not represent the actual flood hazard.

(II) *In which way did the historical transformation of river valleys (e.g., landscape occupation and organization) change the 2021 flood hazard in comparison to past events in this region?*



The cases of the municipalities Schuld and Altenahr (river Ahr) showed that in order to improve flood management practices, it is crucial to better understand the historical land use of the valley such as the reactivation and inundation of the abandoned oxbow lake in Altenburg (Altenahr), which has been urbanized over time. The comparison of the 2021 flood with past events revealed the importance of considering the anthropogenic or natural transformation of the landscape and valleys for flood management. The inclusion of sediments and debris in flood modeling is essential, including new types of industrial large debris, which did not exist in the historical floods analyzed.

(III) *How would the July 2021 precipitation event unfold under different past and future climatic conditions and what implications have these scenarios on flood events?*

A further intensification of such events (both regarding precipitation and resulting floods) is expected with ongoing global warming. Using a series of PGW simulations representing different levels of global warming, the results showed that the precipitation intensity increases in the order of the theoretical Clausius-Clapeyron (CC) scaling with 7 to 9 % per degree warming. The present-day control run already showed an increase of about 11 % in intensity compared to pre-industrial-like conditions, which is equivalent to a doubling of the statistical probability of occurrence. Using the PGW simulations as input for a hydrological model revealed strong non-linear impacts on the hydrological response beyond the CC-scaling.

(IV) *How are precipitation characteristics (e.g., intensity, extent) projected to change under future climate conditions?*

The high-resolution future KIT-KLIWA ensemble also confirms the CC-scaling for the mean and moderate intensities. For more intense precipitation, however, the analyses revealed a super-CC scaling of more than 10 % per degree warming. This implies further intensification of heavy precipitation. Moreover, the area affected by a specific intensity level is expected to grow.

Regarding conclusion (I), different metrics used to classify the July 2021 precipitation event in the historical context considering intensity, duration, and extent revealed that it was among the strongest events in Germany but not unique. The shape and the position of the precipitation areas of these events are emerging from quasi-stationary low-pressure areas over southern or eastern Central Europe (e.g., Stucki et al., 2012; Kelemen et al., 2016) and are often associated with a *Vb* cyclone pathway. This setup mainly favors the occurrence of extremes in eastern and southern Germany, thus, the July 2021 event in western Germany is an exception. Nevertheless, our analyses revealed that the July 2021 precipitation event was less unique in a broader historical and climatological context. Wet soils and local moisture recycling do not seem to be major factors in preconditioning and feeding the event. Here, large-scale mechanisms and the advection of moisture from remote sources were the driving factors (cf. PART1). This was for example also the case in the 29 June 2017 event (Caldas-Alvarez et al., 2022a) where even higher precipitation totals were registered. Nevertheless, the presented results are in line with Schröter et al. (2015), who also found no observed flood-triggering heavy precipitation events on top of a very wet period.

From the hydrological perspective, the July 2021 flood event was exceptional, when (i) compared to long-term gauge recordings from a large region with similar hydroclimate (CReg), (ii) compared to statistical return periods derived from a large set of gauges, and (iii) when compared to long-term historical records at gauge Altenahr (Ahr). While events in the order of mag-





nitude of the 2021 flood were already observed at gauge Altenahr in the past, robust estimation of statistical return periods proved to be difficult and afflicted with large uncertainties. Along both the Ahr and Erft River, the 2021 event exceeded the statistical 100-year return value ($HQ_{100}$) by a factor of up to 7, which is so far beyond observed gauge records that a return pe-
riod estimation is questionable to impossible. The July 2021 flood event, therefore, should be seen as a wake-up call to include additional information as much as possible in the risk assessment and estimation of hydrological and hydraulic design values rather than relying mainly on extreme value statistics based on gauge recordings with limited time coverage. Such additional information could be reconstructed from historical floods or hydrological simulations forced by long-term climate simulations including the effects of climate change as boundary conditions.

In terms of hydro-morphodynamic processes (conclusion II), the analysis of historical events indicated that the current methods for the evaluation of flood hazards do not capture or account for changes in hydro-morphology of river networks such as landscape organization and occupation. These historical changes include also the influence of human activities as constructors of the landscape and river morphology, a role which was considerably intensified in the last century. The consideration of the history of the landscape (based on images and reports and on paleogeography investigation) reveals additional areas prone to
inundation or damage by sediment and debris, such as the abandoned oxbow lake in Altenburg (Altenahr) or the Ahr River bend in Schuld. Furthermore, the consideration of sediments and large debris and associated geomorphological processes is essential to fully capture the flood hazard (Dietze et al., 2022). Several natural and anthropogenic aspects that condition the evolution of the landscape are climate-driven as well, which means that the determination of flood hazards and risk is a highly complex task in the context of global warming.

Regarding conclusion (III), the July 2021 event was re-simulated with the WRF model by altering the mean temperature state from –2 K to +4 K and keeping the relative humidity constant. The control run ($\pm\,0\,\text{K}$) uses the present-day conditions, which represent a global warming level of already +1.09 K according to IPCC (2022). It could be demonstrated that the event precipitation follows the theoretical CC-scaling of about 7 to 9 % increase per degree warming, which is in line with findings by Trenberth et al. (2003) or O'Gorman (2015). A higher CC-scaling is found for the larger LReg underlining the unusual nature
of the July 2021 event in terms of its spatial extent. Putting the spatial mean precipitation from the PGW simulations into the statistical context of LAERTES-EU, the increase in precipitation is equivalent to a doubling of the return period for both LReg and SReg from 5 to 10 years, which is equivalent to a doubling in the probability of occurrence compared to pre-industrial-like conditions. The +2 K simulation predicts an increase in precipitation of 11 % for SReg, which means a further doubling of the return period to 20 years. For LReg, an increase of 18 % is simulated leading to a dramatic increase of the return period
to 200 years. For gauge Altenahr (Ahr), the PGWwarm (PGWcold) scenario with an 18 % increase (11 % decrease) of rainfall led to a 39 % increase (19 % decrease) of the flood peak. This emphasizes the non-linear relationship between meteorological drivers and hydrological response with the potential to magnify hazards related to climate change along the meteorological-hydrological-morphological process chain.

     Regarding conclusion (IV), the convection-permitting ensemble simulations of KIT-KLIWA over the study area agree on an
intensification of extreme precipitation in a warmer climate. Precipitation at a given return period increases across all duration levels studied. The analyses confirm an increase in moderate precipitation intensities following the CC-scaling, but also indicate





an intensification above the CC-scaling (super-CC scaling) of more than 10 % per degree warming for the highest return periods consistent with Feldmann et al. (2013) or Lenderink et al. (2021). Furthermore, the spatial extent of events is expected to grow. Both trends lead to a general increase in the probability that a location is affected by precipitation extremes. Especially for sensitive areas, such as the steep terrain along the Ahr Valley, or generally, the accumulated effect of such extreme events along river reaches, we thus expect an increased hazard potential in a warmer climate. However, the results show no clear signal regarding the frequency of extreme events. Moreover, the study is limited by the number of 30 simulation years for each of the four ensemble members, leading to large uncertainty with increasing return intervals. Therefore, further analyses with more data are needed.

To summarize, precipitation, especially heavy precipitation, remains a challenging task for both observational analyses and statistics, as well as for model simulations and future projections (cf. Stocker (2014); IPCC (2021)). Observational records in most cases are still too short to fully capture the intensity and frequency of extremes that are basically possible. Synthetically extending these records with model simulations also has limitations due to the complexity of precipitation formation, which leads to distinct biases in the simulations. These shortcomings directly propagate into hydrological models and resulting discharge statistics such as the $HQ_{100}$. Furthermore, discharge records are likewise available for short periods only, requiring statistical extrapolations for higher return periods with related uncertainties. However, official guidelines and regulations for flood risk assessments and mitigation rely solely on these observation-based metrics. The presented analysis could show to some extent that consideration of historical records such as photos or written chronicles in the reconstruction of past flood events can reduce the uncertainty of metrics such as the $HQ_{100}$. The general inclusion of such sources would significantly improve the evaluation of the potential hazard. Within the scope of ongoing climate change and the expected further increasing precipitation intensities, this becomes increasingly relevant. Another complement is the forcing of the existing hydrological water balance models with precipitation data from the high-resolution future scenarios to investigate probable future changes in discharge statistics, which can then be used for mitigation purposes. With the recently ongoing development in climate research towards higher-resolution, convection-permitting simulations in the near future (e.g., within the BMBF project "NUKLEUS" for Germany), it is expected that the precipitation statistics will become more robust. We anticipate that this will increase the potential for spatial information to be better represented.

*Data availability.* HYRAS-DE, RADOLAN, and KOSTRA are freely available for research at the Open Data Portal of the German Weather Service DWD (https://opendata.dwd.de, last access: 8 June 2022). HYRAS data can be requested at DWD for research and education purposes. Gauging data are provided by the Global Runoff Data Centre (GRDC) and are freely available at the GRDC portal (https://www.bafg.de/GRDC/, last access 22 May 2022). Regionalized flood information for Baden-Württemberg (BW-Abfluss) is freely available at the LUBW portal (https://udo.lubw.baden-wuerttemberg.de/projekte/, last access 22 May 2022). River gauge data are available on request from the responsible water authority: Water administration of Rhineland-Palatinate (https://www.lfu.rlp.de, last access: 9 May 2022) for gauges Müsch, Altenahr, Jünkerath, Kordel, Prüm 2, and Prümzurlay; Erftverband (https://www.erftverband.de, last access: 9 May 2022) for gauges Schönau and Bliesheim; Wupperverband (https://www.wupperverband.de, last access: 9 May 2022) for gauges Hückeswagen and Opladen. The LAR-SIM hydrological simulations based on the PGW studies are available upon request from the Water administration of Rhineland-Palatinate.



The WRF model code can be obtained from https://github.com/wrf-model/WRF/releases (last access: 13 July 2022). ERA5-forcing data can be downloaded from the Copernicus Climate Change Service (C3S) Climate Date Store (https://cds.climate.copernicus.eu/#!/search?text=ERA5&type=dataset, last access: 13 July 2022). LAERTES-EU, KIT-KLIWA, and the PGW simulation data can be requested from the authors. It is planned to provide parts via the German Climate Computing Center (DKRZ).

*Author contributions.* All KIT authors jointly designed the research questions of the study, continuously discussed the results, and wrote and reviewed the text passages. BM and ACA were responsible for the analysis of historical precipitation events; FE worked on the statistical analysis of the LAERTES-EU data. UE, FS, and JD were responsible for the hydrological analyses (including collection and description of the data). MF was responsible for the hydro-morphological part. MH and HF took care of the future climate projection; PL performed and analyzed the PGW simulations.

*Competing interests.* One of the coauthors (JGP) is a member of the editorial board of *Natural Hazards and Earth System Sciences*. The peer-review process was guided by an independent editor, and the authors have also no other competing interests to declare.

*Acknowledgements.* This study is the result of an interdisciplinary collaboration at the Karlsruhe Institute of Technology (KIT), originating from a Forensic Disaster Analyses (FDA) on the flood of July 2021 conducted by the Center for Disaster Management and Risk Reduction Technology (CEDIM) in summer 2021. CEDIM is a cross-disciplinary research center in the field of disasters, risks, and security at KIT
funded by the KIT and the research program "Changing Earth – Sustaining our Future" in the Helmholtz research field "Earth and Environment". Several authors acknowledge partial funding from BMBF "ClimXtreme Module A" (01LP1901A), BMBF "RegIKlim-NUKLEUS" (01LR2002B), BMBF "RegIKlim-ISAP" (01LR2007B) and DFG "Waves to Weather" TRR 165. Additionally, PL has been supported by the Helmholtz Association (Climate Initiative REKLIM grant) and JQ's contribution was funded by the Young Investigator Group "Sub-seasonal Predictability: Understanding the Role of Diabatic Outflow" (SPREADOUT, grant VH-NG-1243). JGP thanks the AXA Research
Fund for support (https://axa-research.org/en/project/joaquim-pinto, last access: 9 May 2022). The authors thank the Deutscher Wetterdienst (DWD) for providing the HYRAS, HYRAS-DE, KOSTRA, and RADOLAN data sets, and the German Climate Computation Center (DKRZ, Hamburg) for providing computing and storage resources under the projects 105 and 983. The authors thank the Water administration of Rhineland-Palatinate and Baden-Württemberg, the Erftverband, the Wupperverband, T. Roggenkamp from the University of Bonn, Germany, and the Global Runoff Data Center (GRDC), Koblenz (Germany), for providing valuable hydrological observational data and
related evaluations. The authors thank the Water administration of Rhineland-Palatinate and HYDRON GmbH, Karlsruhe (Germany) for carrying out all hydrological PGW simulations with LARSIM. Special thank goes to the *Kreisarchiv Ahrweiler* and Dieter Könnes for photo credits. Finally, we thank the open-access publishing fund of KIT.



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
