# Peer review of "A multi-disciplinary analysis of the exceptional flood event of July 2021 in central Europe. Part 2: Historical context and relation to climate change"

_Natural Hazards and Earth System Sciences, 2022_

## Author Response (AR1)

**REVIEWER #1**

First, we would like to thank the reviewer for her/his insightful comments, which have greatly contributed to improving the text. In making corrections, we have tried to follow as closely as possible the suggestions made.

The manuscript by Ludwig et al. is part II of two papers discussing an exceptional flood event in 2021 in Central Europe. Part II's goals are manyfold: it discusses precipitation and discharge records and valley morphology in historical context and potential future flood events in storyline and in projection approaches. The paper is interesting to read. But it is also very long, sometimes a bit difficult to read, and it is not obvious what is the new take-home message. The manuscript seems to have taken advantage of perspectives, methods, and simulations available at the Karlsruhe Institute of Technology and merged them into one text. This is my main concern. The manuscript should better explain

(a) why precipitation, discharge, and valley morphology are discussed in one paper and what is to be learned from this (it discusses these aspects more or less independently),

A: These are all important points/ingredients to better understand the resulting flood event in 2021. One of the main aims of this study is to put the event in a historical context and to decipher the connections between extreme precipitation and discharge and the role of the specific morphology in the Ahr valley. Based on our analysis, we found that although precipitation was strong during the 2021 event, even stronger events are possible based on the LAERTES-EU ensemble. The discharge of course depends much on the exact location of the precipitation, particularly when considering very narrow valleys like the Ahr and its relatively small catchment area. Although the precipitation was not record breaking during this event, discharges where at the upper limit of the observations, also fostered by the valley morphology. Thus, a more severe precipitation event (whether under present or future climate conditions) at the same location could potentially lead to a much more severe discharge (flood) event. In the revised version, we tried to link and connect the different disciplines better.

(b) why the future climate is discussed with PGW and with high-resolution projections (KIT-KLIWA), but the discussion of advantages and disadvantages concerning telling something about floods in Central Europe is lacking ("KIT-KLIWA ... confirms the CC-scaling" is about all; are the PGW experiments needed if anyway only the well-known C-rates are applied to obs. data in runoff simulations?), and

A: In the PGW experiments, we only investigated the influence of thermodynamic changes on the specific 2021 event. With the KIT-KLIWA ensemble, we consider the full climate change signal under specific global warming levels, including changes in atmospheric dynamics that are neglected by the PGW approach. Thus, the results are also a proof of concept, that such PGW studies might be an alternative way to discuss the impact of global warming on extreme (precipitation) events. However, we have modified the text in the introduction, the description of the PGW experiments (subsection 2.2.2 Pseudo-global-warming experiments with WRF) and in the conclusion part of the revised manuscript.

(c) what can be learned for other catchments worldwide (concerning climate change, land use, methods or whatever - why a scientific paper and not a report?).

A: First of all, the main goal of the two companion papers PART1 (Mohr et al., 2022) and PART2 (this manuscript) is to provide a scientifically sound and comprehensive account of the 2021 event in all its facets (see PART1). But it also goes clearly beyond a mere report of the event by putting it in PART2 into the perspective of (i) historical events, and (ii) climate change. We think this justifies submission as a research paper. Nevertheless, it is a valid question to ask – going beyond the target region and the 2021 event itself - what can be learned for other catchments worldwide. We suggest there are the following three points related to the estimation of hydrological extreme values in a changing climate: (i) Beware of extrapolating gauged-based extreme values too far (e.g., estimating 10,000-year floods from less than 100 years of gauge observations), (ii) CC-scaling takes place, and should be taken into account, (iii) CC-scaling might be further amplified by the non-linear rainfall-runoff transformation. All of these points are already discussed in the discussion and conclusions section 5. We have added to the same section in the revised version of the manuscript a sentence summarizing the generalizable hydrological implications of our study.

*Mohr, S., Ehret, U., Kunz, M., Ludwig, P., Caldas-Alvarez, A., Daniell, J. E., Ehmele, F., Feldmann, H., Franca, M. J., Gattke, C., Hundhausen, M., Knippertz, P., Küpfer, K., Mühr, B., Pinto, J. G., Quinting, J., Schäfer, A. M., Scheibel, M., Seidel, F., and Wisotzky, C.: A multidisciplinary analysis of the exceptional flood event of July 2021 in central Europe. Part 1: Event description and analysis, Nat. Hazards Earth Syst. Sci. Discuss. [preprint], https://doi.org/10.5194/nhess-2022-137, in review, 2022.*

**In some places the wording should be clarified, or minor mistakes corrected:**

line 16: "scales to first order"? Does it mean it is between 1/10 and 10 times the CC-rate?

A: We removed "*first order*" as precipitation scales with CC-rate

l32: "Europe in the last half-century" - last 50 years?

A: Changed as suggested

l35-39/40 could be deleted.

A: Deleted as suggested

l55: perhaps "A key aspect is a deeper analysis of the 2021 flood event taking a ... perspective"?

A: Changed to: "*A key aspect is a deeper analysis of the 2021 flood event based on a long-term climatological perspective.*"

l73: "GCMs (usually 100 to 200 km)" should be more specific. HiResMIP or paleMIP models have very different grid spacings.

A: In the revised versions, we've added information of the range of grid spacing of the commonly used ScenarioMIP runs in the introduction.

l94: PGW is also imprinted on the lateral boundary forcings, I assume?

A: '*lateral boundary*' forcing has been added

l171: "Therefore, ..." - please explain

A: This sentence has been deleted, as it is more a less a doubling of information already provided before.

l210 vs l747: GWL of present-day is 0.46 K or 1.09 K

A: The 0.46K corresponds to the GWL of the reference period (1971-2000) in comparison with pre-industrial conditions based on different observational datasets (Vautard et al., 2014). As the regional model simulations only start after 1950, the period 1971-2000 is considered as the reference period, thus already featuring a GWL of 0.46K with respect to the pre-industrial period. This means that climate change signals of the RCM simulations under GWL2 (with respect to the pre-industrial period) correspond to a +1.54K global warming with respect to the reference period (1971-2000). The 1.09K is the current (2011-2020) GWL based on observation with respect to the pre-industrial period (IPCC, 2021). To avoid any confusion, we deleted the 0.46K in line 210 and only refer to the method applied by Teichmann et al. (2018).

*Vautard R. et al. (2014) The European climate under a 2 °C global warming. Environ Res Lett 9:034006*

l265: It should be better explained why you can assume that the LAERTES-EU data provides independent 12500 years of data?

A: As presented in the early years of operational numerical weather forecasts by, for example, Lorenz (1982) or Dalcher and Kalnay (1987), and recently by Fuging et al (2019), the intrinsic predictability of the atmosphere has a limit of about 12-15 days due to the chaotic nature of the atmospheric system. The largest part of LAERTES-EU (namely data blocks 2 and 4) is a so-called forecast ensemble which means, each simulation is initialized ones and then run free for 10 years. Therefore, it can be treated the same way as a numerical weather forecast but on a 10-year (decadal) scale. Furthermore, the simulations were initialized on a day in November but within LAERTES-EU only data from 1 January are used. So, we already have a spin-up phase of about 2 months which is not considered in the ensemble. As mentioned above, after about 2 weeks there is hardly any dependency between the single members of the ensemble anymore, so using a 2-month spin-up, we can assume that LAERTES-EU provides independent 12,500 years of data. We have added a comment to the manuscript related to this topic.

*Lorenz (1982): Atmospheric predictability experiments with a large numerical model. Tellus A, 34, 505–513, https://doi.org/10.3402/tellusa.v34i6.10836.*

*Dalcher and Kalnay (1987): Error growth and predictability in operational ECMWF forecasts. Tellus A, 39, 474–491, https://doi.org/10.3402/tellusa.v39i5.11774*

*Fuqing et al. (2019): What Is the Predictability Limit of Midlatitude Weather? J. Atmos. Sci., 76 (4):1077–1091, 10.1175/JAS-D-18-0269.1*

l298: "in total" - delete?

A: Deleted as suggested

Fig 3: bubble 13 cannot be seen.

A: We agree that bubble 13 is very tiny; we adjusted the range represented by the bubbles so that the smallest value of the data set still has a recognizable size. Furthermore, as suggested by Reviewer 2, we changed from PSI to HPEcrit related quantities in Fig.3 with the bubbles now showing the NAPI, and the x- and y-axis displaying maximum grid point and spatial mean precipitation, respectively. We have adjusted the text accordingly.

l372: "However ..." - this is a statement for the conclusion

A: We agree with the reviewer, that this statement should also appear in the conclusion. Thus, we shifted this paragraph to the conclusions in the revised version.

l655: "The second type ... scaring ... changing the boundaries ...". Please, reformulate.

A: The sentence was in fact convoluted. As the whole subsection was considerably revised, the sentence has been removed in the revised version.

l703: "precipitation intensities" or do you mean the daily mean amounts discussed above?

A: Yes, we are referring to the daily mean amount discussed above. This has been changed in the revised version.

l722: "... found no observed ..." - observed but not found?

A: We agree, this sentence reads strange. We have reformulated it in the revised version.

l772: "basically" - reformulate, please

A: Changed to "*physically*"

l786: here you refer to the spatial information in the PGWs not used in the hydrological modelling? In what respect shall the new simulations help? In understanding the 2021 event?

A: Basic idea behind the PGW approach is to extract the pure thermodynamical effect of climate change on extreme precipitation by nudging the atmospheric dynamics to follow the present-day state. However, due to the model characteristics, parameterizations, etc., there are little

deviations in the spatial structure of the precipitation field which would further propagate into the hydrological model. The intention of the presented analysis was to extract the pure hydrological response to changed atmospheric conditions. Therefore, we estimated the scaling factors for -1K and +2K and applied it on the "best-available" precipitation data for which the flood peak and course were represented best. Hence, conclusions can be drawn in how far climate change already influenced such peak discharges and to which extent it will change in the future. We agree that the last sentence is misleading and we changed it accordingly. Furthermore, we reformulated the beginning of Sect. 4.1.2 (see also replies to Reviewer 2). As we mainly focus on the -1K, control run, and +2K simulations, we also introduced a new naming of the PGW simulations for a better readability. The -1K simulation is declared as PGWcold, the present-day control run as PGWcontrol, and the +2K simulation as PGWwarm.

**REVIEWER #2**

The presented manuscript characterizes the extreme flood event in July 2021 in western Germany from meteorological, hydrological and geomorphological perspectives. The manuscript is a companion manuscript to Mohr et al. (2022) and focusses on putting the above-mentioned event characteristics into historical perspective. Furthermore, modelling experiments following pseudo-global warming storyline approach were performed with a climate model at convection permitting resolution for scenarios within -2K to +4(+3)K range. The resulting ensemble was used to run a hydrological catchment model to estimate the impact on flood flows. Additionally, the hydro-morphological changes and their impact on flood hazard are discussed.

The manuscript is well-written and mostly well structured. However, it is not concise and some excursions can be significantly shortened as proposed below. The analysis of event severity in meteorological and hydrological terms is not only nicely placed into historical context, but due to selection and analysis of various regions, the spatio-temporal perspective of the July 2021 is well elucidated. The use of LAERTES-EU ensemble and two complementing approaches (PGW and scenario ensemble) to assess the effect of climate change on meteorological and hydrological hazard strongly enrich the presentation. I consider the manuscript to be a valuable contribution to the analyses of 2021 flood event and a good fit to NHESS journal. Said that, the current presentation needs significant revision to address several mostly structural issues and a number of minor issues related to formulations. At several occasions clarity is lacking. I therefore suggest major revision.

With kind regards,

Sergiy Vorogushyn

A: First, we would like to thank Sergiy Vorogushyn for his insightful comments, which have greatly contributed to improving the text. In making corrections, we have tried to follow as closely as possible the suggestions made.

**Major comments:**

L105ff: I feel research questions III and IV are very similar and refer to the potential characteristics of precipitation under future climate conditions and their implications for flood events. Consider merging these two questions.

A: We agree with the reviewer, that both RQ III and IV are related to climate change. However, for RQ III we focus specifically on the Ahr event by a storyline approach, while for RQ IV, we use the traditional probabilistic way to detect climate change signals based on an RCM ensemble for a larger region. We would like to keep both questions, as e.g., the results of the discharge modeling refer specifically to RQ III. For clarification, we adjusted RQ III and IV by stronger indicating that RQ III is related to the specific extreme event and how it would unfold in different climates, and that RQ IV is related to the general evolution of precipitation within climate change. Furthermore, we added a brief discussion of both approaches (storylines/PGW and classical climate projections) in the introduction (cf. Comment on Reviewer #1).

L170-174: At this stage it is not quite clear what exactly you intend to do, i.e. what you mean by "broader classification". Some more specific goal setting would be helpful.

A: We agree and have added to the revised version of the manuscript, at the beginning of Sect. 2.1.2, a sentence explaining our objectives, before introducing the data sets required to achieve them.

L274-275: The definition of HPEcrit should be more precise. Do you first identify the exceedance of 50-year return period on the KOSTRA 8x8 km grid and then count cells with exceedance if they form a contiguous area above 1000 km2? Do you compute return periods at 1x1 km HYRAS-DE resolution and compare to 8x8 km2 KOSTRA or do you regrid HYRAS-DE to the KOSTRA resolution? Specify details in the manuscript.

A: First, we did a geographical comparison of the KOSTRA cells with the HYRAS cells to get all HYRAS cells within a KOSTRA cell. Second, we take the 50-year return level from KOSTRA and assign it to all HYRAS cells within. Then, we count all contiguous HYRAS cells exceeding the KOSTRA threshold. At last, an event is fulfilling the HPEcrit, when it exceeds 1000km$^2$, so approx. 1000 contiguous HYRAS cells. We reformulated the paragraph for clarification and better understanding.

Related to the previous comment: Is the area (A) in Table S2 the contiguous area of HYRAS-DE grid cells exceeding 50-year return period for 24hour rainfall in comparison to KOSTRA? If so, specify this in caption.

A: Yes, the area A given in Table S2 is the contiguous area within HYRAS-DE exceeding the KOSTRA threshold for a 50-year return period. This has been specified in the caption.

L303-305: First, not all 26 but only 20 events are displayed in Fig. 2 and Fig. S1. I am not sure if you have enough evidence to claim that if heavy rainfall occurs in eastern and southestern Germany during summer it is associated with Vb weather patterns. This is indeed documented in the literature for some of the presented events, like 2002, but I am not aware of such established association with Vb for all of the listed events. This statement should be relaxed and maybe shortened.

A: First, we have added the missing events to the supplementary figure in the revised version. Also, we agree with the reviewer, that heavy rainfall in eastern and southeastern Germany during summer is not necessarily associated with the Vb weather pattern. We have reformulated this paragraph as suggested.

L313-329: this description can be significantly shortened, particularly by focusing on the July 2021 event in relation to a few other events in terms of precipitation location, area, intensities etc. L325-329: this somewhat arbitrary distinction into two types of rainfall field is difficult since the data only covers Germany and not riparian countries. I would omit this. Having read the analysis in section 3.1.1 I question the introduction and use of PSI in the context of this study. It does not bring much additional information, but causes quite some confusion in the results. Yes, it has persistence as additional characteristic, but results in the end in a quite different ranking of events. Since the purpose of the study is not to compare different event indices (and there are a few others out there, like WEI (Müller & Kaspar, 2014) and xWEI (Voit & Heistermann, 2022)), but to put the July 2021 into the historical context, I would suggest to focus only on the HPEcrit index and omit PSI. It would improve the clarity of the manuscript. Figure 2 should then be redesigned for HPEcrit index.

A: Thank you for this comment. As suggested, we have shortened and rearranged Section 3.1.1., also skipping the paragraph on the separation into two types. We agree that the PSI part feels

like a foreign body with no clear added value regarding our research questions. Therefore, we have changed Fig.3 as suggested from PSI to other quantities given in Table S2 that are partly included in HPEcrit as well and adjusted the text accordingly. Furthermore, we have removed the PSI-related columns in Table S2.

Chapter 3.3 needs considerable revision. It starts with putting the July 2021 event into the historical perspective but then loses the focus. I would suggest to focus on general changes in the landscape including (1) urbanization, (2) construction of bridges and transport of infrastructure, (3) land use & agricultural practice and support them with a few examples.

A: We have revised and simplified this Subsection. We didn't necessarily align it according to the topics suggested by the reviewer, but we believe that it is now clearer, more organized and linked to the factual observations, and better streamlined with the objectives of the paper.

Section L510-529 can be omitted. You can use the examples from local places (Altenahr, Schuld, Dernau), but these should be introduced and shown on a map. It is not possible for a reader not being acquainted with the local geography to understand these details. Either refer to figures PART1 or introduce those localities here. In overall, I would significantly shorten the description of those examples of geomorphological long-term changes.

A: During a comprehensive revision of the complete subsection 3.3 taking into account also this comment, large parts of this paragraph were excluded or partly integrated in the conclusions. We also included a reference to figures in PART1 showing the discussed locations on maps.

L548-551: This finding is not surprising if you prescribe CC-scaling for specific humidity of the initial and boundary data (L230-232), is it? Then one should rather discuss the limitation of this setup and not directly confirm it with observational evidence.

A: Climate model projections show that the increase in water vapor leads to robust increases in precipitation extremes everywhere, with a magnitude that varies between 4% and 8% per 1°C of surface warming (see IPCC AR6, Chapter 11, p 1526). However, this change is not transferred 1:1 into precipitation as other dynamical and local drivers are important for precipitation formation and not just moisture content. Thus, there is a relatively wide range of precipitation increase regarding the imprinted thermodynamic forcing. With our PGW simulations, we can estimate the exact precipitation increase per 1K warming precisely and specifically for the 2021 event. However, we agree that some discussion on the limitation of the setup (only changes in thermodynamics, not in the dynamics) is lacking, which has been adjusted in the revised version (additional information on these issues are now provided in the introduction and subsection 2.2.2).

L553 -561: This paragraph should be formulated more precisely. Are you talking here of the average July 2021 precipitation in a specific region? I am then puzzled, how can the return period of this precipitation in a colder climate become smaller (5 years) compared to the warmer climate (20 years). L558: there is no return period of the control run, but of some precipitation amount. Overall, it is difficult to keep track of numbers and their changes for specific regions, please, consider bringing these numbers and regions into one table. Also, the hydrological results (L580-586) can be introduce into this table.

A: We agree that this is a bit confusing and we reformulated this paragraph accordingly. In principle, we calculated the spatial mean daily precipitation totals for the LReg and SReg domain from the PGW control run, the -1K, and the +2K scenario. These values (in mm) are

put into the LAERTES statistics (Fig. 5), which always refer to present-day conditions. This means, for example, that the precipitation total of the present-day PGW control run for the smaller SReg domain has a return period according to present-day LAERTES statistics of about 10 years while the precipitation totals of the PGW -1K simulation would have a return period of 5 years in present-day LAERTES statistics, i.e., when it would occur today, and so on. We also added the missing values and merge all numbers of sections 4.1.1 and 4.1.2 as suggested into a new table where possible.

L586: Can the conclusion about the amplification of hydrological response be supported by the model analysis? You can derive runoff coefficients, share of quick(overland) flow, concentration times. Of course, these are modelled indicators which may or may not reflect the reality, but at least it will explain the model response.

A: It would be really interesting to investigate in more depth the particular hydrological causes for the amplification of the CC-scaling in the hydrological response. However, it is beyond the scope of this work, as we do not have direct access to the Larsim model: the calculations were done by the water authorities of Rhineland-Palatinate. However, in the revised version of the manuscript, we added to Sect. 5 a sentence suggesting this analysis for further research.

L611-612: Why is it so? Can you speculate about a possible reason?

A: The presented relative change estimates are average values over all ensemble members and grid points. We took a more detailed look at it by including the ensemble spread via the interquartile range (25th to 75th percentile) in Table S3. Doing so, it turned out, that the rather small differences between CReg and LReg of about 1% are within the uncertainty range/confidence bounds of the ensemble. Therefore, we removed this sentence and reformulate the paragraph relating to the new additional numbers in Table S3.

Chapter 4.3 feels like a foreign body. It is not really linked to the rest of the manuscript and contains very general discussion not linked to any concrete findings. I suggest to remove it completely, but use some (few) thoughts and literature references in redesigned Chapter 3.3 focusing on geomorphological perspective. But the focus on historical changes should be kept in order to be consistent with the original scope of the manuscript.

A: We swapped the initial Subsection 4.3 for one more simplified and focusing on the potential effects of climate change in hydro-mophodynamic processes supported by additional literature, more overall aligned with the Section.

Discussion and conclusion chapter require considerable revision. It is partly very general and does not distill main findings. L684-721: the attempt to provide answers to the originally formulated research questions does not work in my view. The questions are not narrow enough to be answered in a few sentences, so that they can be understood without further information. A more detailed information is provided below, but it is decoupled from the above answers. I suggest to dissolve this structure and provide the concise discussion of the four questions/issues and related findings in plain text. It would have 4 (or 3) paragraphs (if you merge QIII and QIV) summarizing the discussion.

A: We have restructured the discussion in the revised version with a specific emphasis on merging the initial answers to the research questions and the more detailed information provided later. We hope that the reorganization of the discussion better distills the main findings and provide more direct answers to the research questions than in the original version.

**Minor comments:**

L3: Part 1 – substitute by the proper reference to the (discussion) paper.

A: According to the NHESS guidelines (see below), there should be no references in the abstract. Therefore, we decided to keep "*Part 1*" at this point. The proper reference to the discussion paper is given in the introduction.

Excerpt from NHESS guidelines: "*Reference citations should not be included in this section, unless urgently required, and abbreviations should not be included without explanations.*"

L5 flood hazard

A: Changed as suggested

L7: return values and periods – what is the difference between the two?

A: Return period refers to the time interval in which a specific value occurs or vice versa, return value is the corresponding absolute value of a variable at a specific return period. However, a more accurate expression for "*return value*" is "*return level*". We adjusted the text by using return level and return period, solely, instead if mixing with return value and/or return interval.

L11: hazard assessment of flood risk – reformulate

A: Reformulated to: "... which were not included in the ***flood risk assessment***"

L29: remove 'widespread'

A: Changed as suggested

L31: please, specify that death toll of more than 180 refers to Germany only. The description above was focusing also on non-German part. Also, afterwards economic losses including neighboring countries are mentioned.

A: We have adjusted this accordingly and update the numbers including a new reference:

*Tradowsky et al. (2022): Attribution of heavy rainfall events leading to the severe flooding in Western Europe during July 2021, Clim. Change in revision.*

L44-54 can be omitted.

A: This paragraph has been slightly revised, but we would like to keep this basic information on the event here.

L58: 'in terms of peak discharge'. Actually, the 1910 event peak was about the half of that in 1804 and 2021. I would therefore not speak of 1910 being comparable to 2021.

A: We agree and rephrased this comparison, stating that 2021 was comparable to the 1804 event, and we replaced '***in terms of discharge***' by '***in terms of peak discharge***'

L59: you mean here official estimates by LfU RLP. This should be mentioned.

A: Changed to: 'However, these events were not considered for the *official* estimation of the 100-year return periods of discharge (HQ100) *by the Landesamt für Umwelt (LfU) of RP* as the continuous time series of observations only starts in 1946.'

L60: Vorogushyn et al. (2022) – meanwhile published.

A: Reference has been updated.

L114: three rectangular geographical domains

A: Changed as suggested.

L147ff: It should be noted that all values are rough estimates and are not directly measures due to failure of gauging stations.

A: We agree that many of the streamflow values are based on reconstructions, especially the peak values, but not all of them. In PART1 (Sections. 2.3 and 3.2, and Figure 6), we explain and show which values are based on measurements, and which are based on reconstructions. In the revised version of the manuscript, we have added a related sentence to the first paragraph in Sect. 2.1.2.

L166: However, the estimations of return periods across different federal states are inconsistent – Is this what you want to say?

A: Yes, exactly.

L258-263: Can this be omitted?

A: We agree that this is more or less textbook knowledge. We reduced this paragraph to a high degree just mentioning the most important parts related to the study.

L205 & L228: Here you are talking about the PGW up to +3K. In the abstract up to +4K – correct.

A: Corrected in the abstract; should be +3K.

L247: +2K – GWL2 or +3K – GWL3?

A: Based on IPCC AR6, global temperatures are currently +1.09K (~1K; GWL1) above the pre-industrial reference period. So, a reduction of -1K in the 2021 ERA5 data corresponds to GWL0 (=pre-industrial). An increase of 1K in ERA5 would be equal GWL2, an increase by 2K equals GWL3. We rephrased the sentence in L228f to clarify already here that pre-industrial is the reference period, and we currently observe a global warming of ~1K (GWL1).

We changed L228f: '*The control run (± 0K) uses the present-day conditions, which represent a global warming level of already +1.09K (GWL1) according to IPCC (2022). Thus, a* reduction of 1K would represent temperatures as in the pre-industrial period **(GWL0)**, while an increase of 2K corresponds to GWL3.'

L273: specify here that you focus on daily precipitation totals.

A: Sentence changed to: 'In addition to the antecedent precipitation index (API) used in PART1, two other measures are applied to the HYRAS-DE data for the classification of precipitation events: the first one is the empirical heavy precipitation event criterion $HPE_{crit}$, which combines thresholds for magnitude and extension *based on daily precipitation totals*.'

L298: is illustrated

A: Changed as suggested.

L299: Omit last sentence.

A: Sentence has been deleted as suggested.

L317: awkward sentence – reformulate

A: Due to a more complex revision of the complete section 3.1.1 (see comment above), this sentence was deleted.

L376: do you mean "extent" or rather the location of the precipitation field within CReg?

A: Yes, we mean the location of the precipitation field within CReg. The sentence has been reformulated: 'The results of the spatial analysis are in line with those of PART1 and the findings of the previous section that the *location of the extended precipitation field* of the July 2021 event was special.'

L397: You can easily fit linear models and show how strong the linear relationships are. What is "approximately linear"?

A: We agree that 'approximately linear' is imprecise, but it is not our goal to assess in detail the nature and strength of the (linear) relation between peak streamflow and basin size. Rather, we wanted to describe to the reader that a relationship exists, and that it is broadly linear. In a revised version, we rephrased the sentence to: "For both data sets, a linear dependency between peak streamflow and basin size is visible".

L399: I cannot confirm that only 5 GRDC events exceed the 2021GD. How do you determine this? If you fit a linear model to 2021GD data and even if you shift it to the most outlying point retaining the slope (so, that the other red dots are below), would it be still exceeded by only five GRDC events? Looks like not. But it should be shown. L400-405: these peaks/gauges can be marked in Figure 6.

A: We agree that the selection of the 5 events exceeding the 2021GD events was subjective and based on visually selecting from Fig. 6 the ones that particularly stood out. For a more objective comparison, we now did the following: Divide all streamflow data in Fig. 6 by the respective catchment area for comparability, sort these normalized peak streamflow values by size, and within this sorted set determine the average rank of all 2021GD gauges. Of the overall 152 values, the 2021GD values received the average rank of 22.8, which is considerably larger than the average rank of 152/2 = 76. We added this result to the revised version of the manuscript. For brevity we removed the discussion of the 5 peak flows.

In Figure 6 I miss the peak for Altenahr and Müsch in the 2021GD dataset, ~1000 m3/s and ~500 m3/s, respectively. The red dot at ~500 m3/s is not Müsch but likely Kordel at Kyll with 816 km2 basin area. I still count 10 red points. If two are missing, what are the other two?

A: All 10 gauges are included in Figure 6, but the values are – for comparability with the GRDC data - daily averages instead of hourly values. This is explained in the first sentence of Sect. 3.2.1 and also mentioned in the caption of Fig.6. The values are:

| Gauge name | Maximum mean daily streamflow [m³/s] (rounded) | Basin size [km²] (rounded) |
|---|---|---|
| Müsch | 158 | 353 |
| Altenahr | 442 | 749 |
| Jünkerath | 81 | 175 |
| Kordel | 489 | 816 |
| Prüm2 | 27 | 53 |
| Prümzurlay | 341 | 576 |
| Schönau | 26 | 31 |
| Bliesheim | 268 | 604 |
| Hückeswagen | 108 | 163 |
| Opladen | 402 | 606 |

We have added these values to Table 1 in the revised version and adjusted the text of Section 3.2.1 accordingly. We think this supports Fig. 6 and our storyline.

L418: a few other return periods

A: Changed as suggested

Figure 7: What are orange dots? These are not annual maximum peaks, but a few peaks above a threshold? Please, explain the caption. Hence, the statement in L431-432 is not correct. The HQ100 estimate of 241 m3/s is based on the AMS in the period 1947-2016.

A: The orange dots are the 10 largest floods on record at gauge Altenahr, as provided by the water authority of Rhineland-Palatinate (https://geodaten-wasser.rlp-umwelt.de/prj-wwvauskunft/projects/messstellen/wasserstand/register2.jsp?intern=true&msn=2718040300&pegelname=Altenahr%20&gewaesser=Ahr&dfue=1). We rephrased the figure caption to clarify this.

And indeed, the statement in L431-432 is not correct, we have rephrased it to "Figure 7 shows the available flood events at gauge Altenahr. The events in orange are the ten largest flood events in the gauge recordings starting in 1946. The gauge recordings from 1947 - 2016 were the basis for the HQ100 estimate (241 m3 s−1; Table 1) by the water administration of RP".

L436-437: gauge recordings did not miss the historical peaks, because no recordings were carried out prior to 1946 - reformulate.

A: We agree and reformulated this in the revised version of the manuscript.

L439: Henrichs with one 'n'.

A: Reference has been corrected

L443ff: I suggest a slight reformulation here related to the work of Vorogushyn et al. (2022), where I was a co-author. (1) It was exactly the purpose of the study to show the limitation of the extreme value statistics without considering historical floods, and not to estimate an

unrealistic return period of 10^8. (2) It should be noted that Vorogushyn et al. (2022) did not use 2021 event for fitting the distributions, while Henrichs (2022) did. So, the estimates are not fully comparable.

A: We agree and have reformulated this in the revised version of the manuscript

L468-472: is there photographic evidence or any type of documentation and analysis?

A: As we don't have any own photos, we have added a reference (https://www.rnd.de/panorama/ahrtal-so-geht-es-den-menschen-nach-der-flutkatastrophe-LEXREN3YXKOQEAK2FK4POQULRM.html) to a figure showing all the different kind of debris (caravans, containers, etc…).

L477: this statement requires a reference.

A: We have added the following reference, that also mentioned that the old oxbow lake in Altenburg (Altenahr) was inundated.

Szymczak, S., Backendorf, F., Bott, F., Fricke, K., Junghänel, T., and Walawender, E. (2022): Impacts of Heavy and Persistent Precipita tion Railroad Infrastructure in July 2021: A Case Study from the Ahr Valley, Rhineland-Palatinate, Germany, Atmosphere,13 https://doi.org/10.3390/atmos13071118.

L563-566: This can be omitted.

A: Removed as suggested.

L624: This statement is not precise, unless areal dimensions are not specified. Since thick lines in Figure 12 have different slope, the relative change will be different. Instead of the discussion in L624-629 I would suggest to discuss the mechanisms of increasing precipitation clusters in the model, and support this by literature references analysis observational evidence if exists. The notion that for large clusters model boundaries provide represent a limitation is valuable.

A: We agree that this paragraph was not precise enough. We added a comment on how the areal dimension was specified. As suggested, we changed the discussion of L624-629 towards a more process-related one.

L717: not setup, but weather pattern.

A: Changed as suggested

L716-718: this comes out of blue. At any point in this manuscript it was mentioned that July 2021 was due to Vb weather pattern or its variation.

A: As already discussed in a previous answer, we rephrased the sentences regarding the influence of Vb weather patterns on extreme precipitation in Germany in the revised version.

L723-724: this statement referred to 2013 flood. It gets messy here (L719-723). Please, discuss rather your concrete findings on the uniqueness of precipitation related to Question I here. E.g. one important message: the event was quite exceptional for the region, but not unique in terms

of intensities and extent for a larger area in Germany and so on. In this respect, the discussion of the hydrological perspective (L724-734) is a perfect blue print.

A: This has been reformulated while revising the discussion section. We have adjusted it towards the direction of the hydrological perspective.

L746: +4K or +3K?

A: Corrected to +3K

L750-754: the formulations should be reformulated more precisely. It is not clear what you mean: The increase in precipitation corresponding to increase in return period from 5 to 10 years… (which precipitation, in which scenario? Increase compared to what?). Doubling of the return period from 5 to 10 years actually means reduction of probability of occurrence of this amount and not doubling. It depends on what you mean in relation to what.

A: As described in the answer to one of the major points above, the reference is LAERTES which always refers to present-day conditions. As we focus on daily precipitation totals in this study, precipitation here also refers to daily precipitation spatially averaged either over SReg or LReg. However, you are right that an increase of the return period is a reduction of the probability. This was a mistake and we fixed that. We rewrote this paragraph accordingly for clarification and include the missing information.

L770: precipitation is not a task – reformulate.

A: We agree with the reviewer, it is a challenging task regarding modeling. We have reformulated this sentence in the revised version.

L777-779: Where were photos and chronicles used in the presented manuscript to reduce the uncertainty in HQ100?

A: This was misleading. We did not use photos etc. to reduce the HQ100 uncertainty. It was meant more as a general statement that such sources might be useful in future HQ100 estimates. However, we removed this in the revised version of the manuscript.

**References:**

Mohr, S., Ehret, U., Kunz, M., Ludwig, P., Caldas-Alvarez, A., Daniell, J. E., Ehmele, F., Feldmann, H., Franca, M. J., Gattke, C., Hundhausen, M., Knippertz, P., Küpfer, K., Mühr, B., Pinto, J. G., Quinting, J., Schäfer, A. M., Scheibel, M., Seidel, F., and Wisotzky, C.: A multi-disciplinary analysis of the exceptional flood event of July 2021 in central Europe. Part 1: Event description and analysis, Nat. Hazards Earth Syst. Sci. Discuss. [preprint], https://doi.org/10.5194/nhess-2022-137, in review, 2022.

Müller, M. and Kaspar, M.: Event-adjusted evaluation of weather and climate extremes, Nat. Hazards Earth Syst. Sci., 14, 473–483, https://doi.org/10.5194/nhess-14-473-2014, 2014.

Voit, P. and Heistermann, M.: A new index to quantify the extremeness of precipitation across scales, Nat. Hazards Earth Syst. Sci., 22, 2791–2805, https://doi.org/10.5194/nhess-22-2791-2022, 2022.

Vorogushyn, S., Apel, H., Kemter, M., Thieken, A.H. (2022): Analyse der Hochwassergefahrdung im Ahrtal unter Berücksichtigung historischer Hochwasser – Hydrologie & Wasserbewirtschaftung, 66, (5), 244-254. DOI: 10.5675/HyWa_2021.5_2